# High-level visual prediction errors in early visual cortex

**David Richter** [1,2]*, **Tim C. Kietzmann**[3], **Floris P. de Lange**[1]

**1** Donders Institute for Brain, Cognition and Behaviour, Radboud University Nijmegen, Nijmegen, the Netherlands, **2** Mind, Brain and Behavior Research Center (CIMCYC), University of Granada, Granada, Spain, **3** Institute of Cognitive Science, University of Osnabrück, Osnabrück, Germany

* david.richter.work@gmail.com

**Data Availability Statement:** Data and code are available from the Radboud Data Repository: https://doi.org/10.34973/8e49-2012.

**Funding:** This work was supported by the ERC Consolidator Grant 2020 "Surprise" (Project

## Abstract

Perception is shaped by both incoming sensory input and expectations derived from our prior knowledge. Numerous studies have shown stronger neural activity for surprising inputs, suggestive of predictive processing. However, it is largely unclear what predictions are made across the cortical hierarchy, and therefore what kind of surprise drives this up-regulation of activity. Here, we leveraged fMRI in human volunteers and deep neural network (DNN) models to arbitrate between 2 hypotheses: prediction errors may signal a local mismatch between input and expectation at each level of the cortical hierarchy, or prediction errors may be computed at higher levels and the resulting surprise signal is broadcast to earlier areas in the cortical hierarchy. Our results align with the latter hypothesis. Prediction errors in both low- and high-level visual cortex responded to high-level, but not low-level, visual surprise. This scaling with high-level surprise in early visual cortex strongly diverged from feedforward tuning. Combined, our results suggest that high-level predictions constrain sensory processing in earlier areas, thereby aiding perceptual inference.

## Introduction

Predictive processing theories promise to provide a principled account of cortical computation [1–4]. One critical ingredient of predictive processing is the computation of prediction errors, i.e., the mismatch between prediction, usually thought of as a top-down signal, and bottom-up input. Such prediction errors then serve as input to the next level in the cortical hierarchy. The brain minimizes prediction errors by recurrently updating its predictions. This process enables the formation of a coherent, stable, and efficient representation of the world. Despite variations in specific predictive processing implementations [1–6], the core concept of prediction error computation is ubiquitous and supported by many empirical observations. For example, after visual statistical learning, visual cortex is sensitive to the likelihood of an object's appearance. In particular, activity throughout the ventral visual stream has been shown to be attenuated to expected compared to unexpected appearances of the same stimuli [7–11]. This attenuation, also known as expectation suppression, has been observed across different species and modalities [12–14] and occurs also when predictions and stimuli are task-irrelevant [15–18]. Combined, expectation suppression has frequently been interpreted in the context of

101000942) awarded to FPdL, the ERC Starting Grant "TIME" (Project 101039524) awarded to TCK, and the Marie Skłodowska-Curie Grant "PreVision" (Project 101147241) awarded to DR. The funders had no role in study design, data collection and analysis, decision to publish, or preparation of the manuscript.

**Competing interests:** The authors have declared that no competing interests exist.

**Abbreviations:** CSF, cerebrospinal fluid; DNN, deep neural network; EVC, early visual cortex; FD, framewise displacement; GLM, general linear model; GM, gray matter; HVC, higher visual cortex; INU, intensity non-uniformity; ISI, interstimulus interval; ITI, intertrial interval; LOC, lateral occipital complex; RDM, representational dissimilarity matrix; ROI, region of interest; RSA, representational similarity analysis; RT, reaction time; SBRef, single-band reference; SVM, support vector machine; TPM, transitional probability matrix; tSNR, temporal signal to noise ratio; V1, primary visual cortex; VIF, variance inflation factor; WM, white matter.

predictive processing as reflecting larger prediction errors for unexpected stimuli, and thus taken as crucial evidence that perception fundamentally relies on prediction [12,14].

If prediction and prediction error computations underlie perceptual inference, as suggested by predictive processing theories, we can stipulate that cortical predictions and the associated prediction error signatures must reflect stimulus features that are represented in the respective cortical area. For example, prediction errors in primary visual cortex (V1) may signal deviations from expectation in terms of simple features such as stimulus orientation, edges, and contrasts —i.e., visual features that V1 neurons are tuned to [19]. On the other hand, prediction errors in higher visual areas (HVC), for instance in fusiform gyrus, may reflect more complex high-level visual features, such as object identities, spatial relationships between object parts and more abstract concepts such as faces, commonly represented in those areas [20–22]. This account suggests that prediction errors mirror local feature tuning, unique to each visual cortical area. While some studies have provided indirect support for the feature specificity of sensory prediction errors by investigating tuning specific modulations [9,23,24], little evidence directly shows which visual feature surprise, if in fact any, is reflected in visual prediction errors.

In contrast to local feature tuning, prediction error tuning may be inherited top-down. Top-down inheritance is in line with hierarchical predictive processing, because predictions are proposed to be relayed top-down from higher to lower visual areas [2], and thus lower visual areas may come to reflect tuning properties of higher areas in predictive contexts due to the top-down prediction signals. Empirical support for this notion has been obtained in the macaque face processing system. For example, Schwiedrzik and Freiwald [25] showed increased neural activity in ML, a lower-level area in the macaque face processing hierarchy that is not tuned to identity, when monkeys viewed face stimuli that were surprising in terms of identity. These results could be explained by feedback signals from higher-order identity-tuned cells in the inferior temporal cortex. Whether a similar principle of top-down prediction error tuning inheritance applies across species, and to stimuli outside the narrow domain of face processing, and across the visual hierarchy remains unknown.

Given that prediction error computation is a core mechanism of predictive processing, it is crucial to characterize what kind of visual surprise is tracked by the visual system. Here, we aimed to close this gap by exploring features reflected in the visual surprise response after statistical learning. Specifically, we asked (1) whether prediction errors come to reflect any visual feature tuning in predictive contexts, and if so, whether (2) this tuning is in line with the local visual features or inherited top-down. To do so, we exposed human volunteers to images that were either expected or unexpected in terms of their identity, given a preceding cue, while recording whole-brain fMRI. To quantify visual feature surprise across multiple levels of description, we used representational dissimilarity metrics derived from a visual deep neural network (DNN). Our results demonstrate that neural responses, and the fidelity of visual representations, across multiple visual cortical areas monotonically increased with how visually dissimilar a surprising object was compared to the expected object. Crucially, high-level visual dissimilarity accounted for the surprise induced increase of neural responses, including in the earliest visual cortical area, V1. Prediction errors thus appear to reflect surprise primarily in terms of high-level visual features, demonstrating that earlier visual areas inherit feature tuning usually associated with higher visual areas in predictive contexts, presumably due to feedback signals.

## Results

Human volunteers ($n$ = 33) viewed images that could contain either animate or inanimate entities. Each image was preceded by a letter cue that probabilistically predicted the identity of

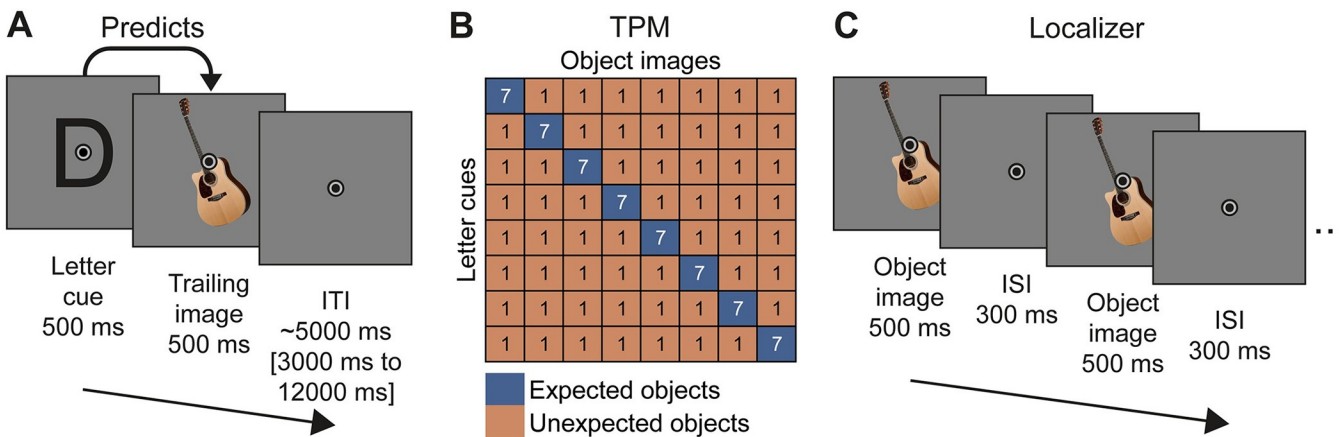

**Fig 1. Paradigm.** **(A)** A single trial, showing a letter cue (500 ms) followed by an image (500 ms) and a variable ITI (approximately 5,000 ms). The image was expected or unexpected given the preceding letter cue. Participants responded by button press to the images, indicating whether the entity in the image was animate or inanimate. **(B)** TPM determining the associations between cues and images. Each of the 8 images was associated with one of the 8 letter cues. The expected image was 7 times more likely to appear than any other image given its cue. Numbers in each cell indicate the number of trials per run. The specific cue-image associations were randomized and differed between participants. Moreover, the set of 8 images also varied for different participants. **(C)** Two cycles of a localizer trial. During the localizer, one image was presented repeatedly (500 ms on, 300 ms off) for 12,000 ms. The identity of the images was not predictable. Participants responded to a high brightness version of the images, which was shown once during each trial for one cycle. ISI, interstimulus interval; ITI, intertrial interval; TPM, transitional probability matrix.

the image. A trial is depicted in Fig 1A. The expected image was seven times more likely to follow its associated letter cue compared to each of the 7 unexpected images (the transitional probability matrix (TPM) is depicted in Fig 1B). The same images appeared both as expected and unexpected stimuli, with the expectation status only contingent on the cue after which the image appeared. Participants were tasked to classify the content of the image as animate or inanimate. Further details are in the Materials and methods: Stimuli and experimental paradigm section.

### Behavioral facilitation by valid prediction

Participants learned and used the underlying statistical regularities to predict inputs thereby facilitating performance (Fig 2), in terms of response times (RT; $F_{(1.4,43.3)} = 23.5$, $p < 0.001$, $\eta_p^2 = 0.42$) and response accuracy ($F_{(1.4,46.3)} = 7.8$, $p = 0.003$, $\eta_p^2 = 0.20$). Specifically, RTs to expected images (501 ms) were faster compared to unexpected images requiring the same button press (509 ms; $t_{(32)} = 2.41$, $p = 0.019$, $d_z = 0.14$) and different button press (524 ms; $t_{(32)} = 6.77$, $p < 0.001$, $d_z = 0.39$). The former contrast demonstrates that surprising stimuli resulted in slower responses even when the stimulus required the same response (i.e., was of the same animacy category) as the expected input. Additionally, adjusting responses resulted in even slower RTs (unexpected same versus unexpected different: $t_{(32)} = 4.36$, $p < 0.001$, $d_z = 0.25$). Response accuracy only showed a reliable decrements when unexpected stimuli required a different response, thus likely reflecting response errors due to invalid prediction (expected versus unexpected different: $t_{(32)} = 3.75$, $p = 0.001$, $d_z = 0.61$; expected versus unexpected same: $t_{(32)} = 0.79$, $p = 0.43$, $d_z = 0.13$; unexpected same versus unexpected different: $t_{(32)} = 2.97$, $p = 0.008$, $d_z = 0.48$). Overall behavioral facilitation thus demonstrated that participants benefitted from prediction. Moreover, accuracy in general was very high (>95%), indicating effective task compliance.

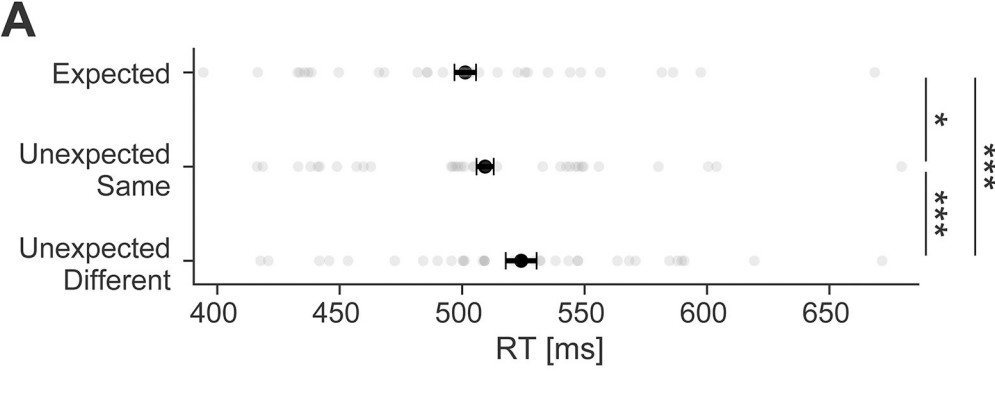

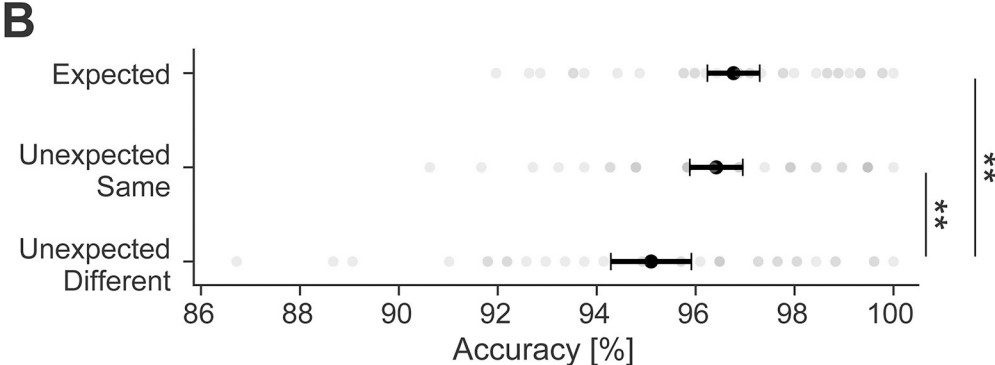

**Fig 2. Behavioral facilitation due to prediction. (A)** RTs to expected objects were faster compared to unexpected images requiring the same or different response as the expected stimulus. **(B)** Accuracy of responses was high overall, but responses were more accurate to expected compared to unexpected images requiring a different response than the expected stimulus. Error bars indicate the 95% within-subject confidence intervals. *** $p < 0.001$, ** $p < 0.01$, * $p < 0.05$. Data and code that support these findings are available at: https://doi.org/10.34973/8e49-2012. RT, reaction time.

## Deep neural network models mirror a gradient from low-to-high level visual features in visual cortex

While (visual) DNNs have been used to successfully explain a variety of neural data [26], we first ensured that the specific feature models utilized here were able to explain visual responses in our data in a prediction-free context. Specifically, we extracted the correlation distance between images from an implementation of AlexNet trained on ecoset [27]. Then, we performed representational similarity analysis (RSA) using the representational distances derived from the DNN layers and from the fMRI data obtained during localizer runs. During these runs, each image was presented in isolation, without preceding cue and without predictive associations between the images. To assess possible contributions of all DNN layers, we repeated this analysis using representational dissimilarity matrices (RDMs) from each of the DNN layers separately. Finally, for each voxel (sphere searchlight) we determined the best DNN layer for explaining neural variance based on the RSA results (correlation coefficient).

Results (Fig 3A), showed a gradient from early to high visual cortex with the corresponding early to late DNN layers best explaining neural variance. Specifically, early visual cortex (EVC) responses were best explained by early layers of the DNN, intermediate visual areas in LOC by early to intermediate layers, and HVC responses, for example, in the fusiform gyrus, were best explained by intermediate to late layers of the DNN. These outcomes affirm previous reports

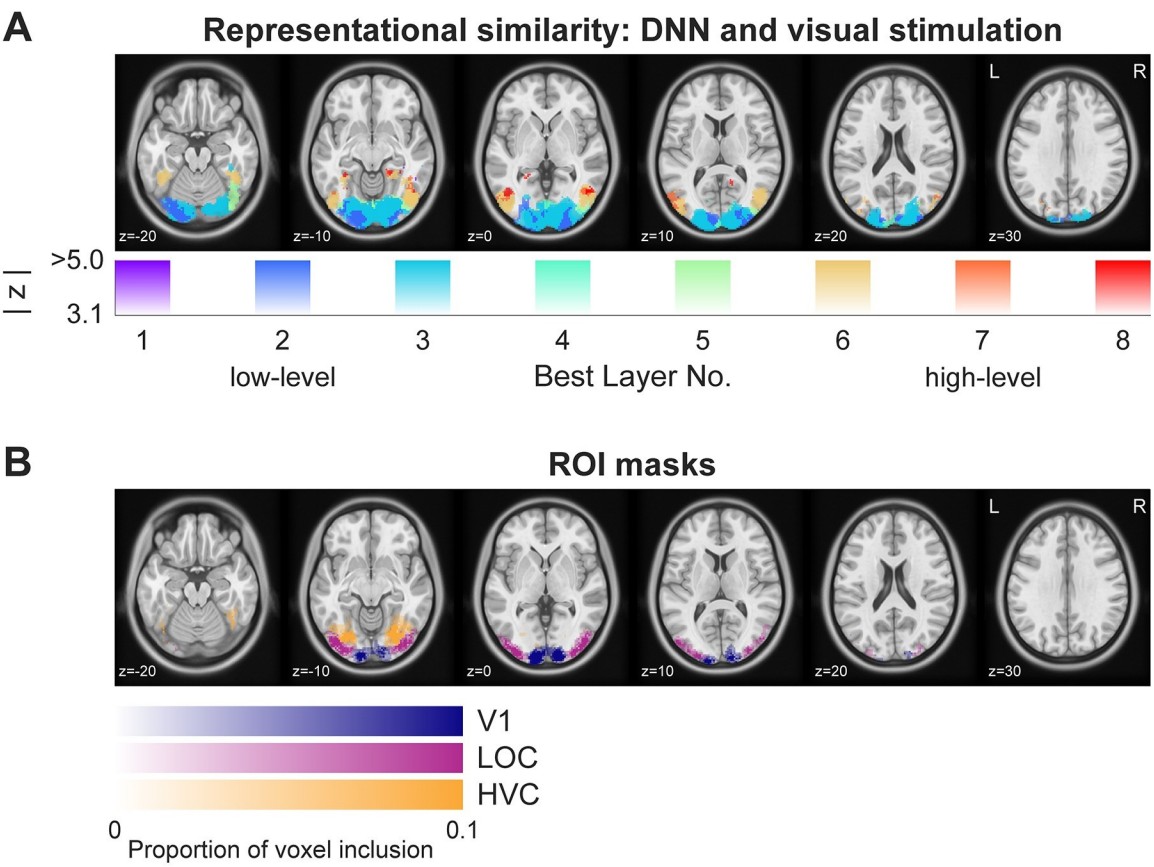

**Fig 3. Convergence of DNN and visual cortical representations showing a gradient from low- to high-level features in prediction-free contexts. (A)** RSA of visual responses shows that (feedforward) visual responses during the prediction-free localizer were best explained by a gradient of low-level to high-level visual features going up the ventral visual hierarchy. EVC responses aligned more closely with early DNN layers, indicative of low-level visual feature processing (depicted in cold colors: purple to blue). HVC areas, like the fusiform gyrus, show a greater correlation with late DNN layers, representing high-level visual feature processing (illustrated in warm colors: yellow to red). Analysis was masked to visual cortex and thresholded at z > 3.1 ($p < 0.001$, uncorrected) of the RSA. **(B)** ROI masks, depicting voxels included in the anatomically and functionally defined masks. Color indicates the ROI: Blue = V1, Purple = LOC, Orange = HVC. Opacity indicates the proportion of participants whose individual masks included the voxel. For visualization, full opacity corresponds to a proportion of 0.1, with voxel inclusion proportions reaching up to ~0.7. Data and code that support these findings are available at: https://doi.org/10.34973/8e49-2012. DNN, deep neural network; EVC, early visual cortex; HVC, higher visual cortex; ROI, region of interest; RSA, representational similarity analysis; V1, primary visual cortex.

[27–31] and validate that our visual feature models, including the low-level (layer 2) and high-level visual model (layer 8), accounted for cortical variance elicited by visual stimulation in an expected pattern of a low-to-high level gradient. In addition, we explored representations of layer 2 and 8 using feature visualization and decoding techniques to further support our assumption that these layer representations serve as low-level and high-level visual feature spaces respectively. For details, see S1 Fig. In brief, results showed that layer 2 represents low-level visual information, such as Gabor-like orientation features, while layer 8 has more complex texture-like [32] and categorical tuning.

### Prediction errors scale with high-level, but not low-level visual feature dissimilarity

Next, we turned our attention to the nature of prediction errors, asking which visual features, if any, they reflect across multiple regions along the ventral visual hierarchy. Fig 4 illustrates

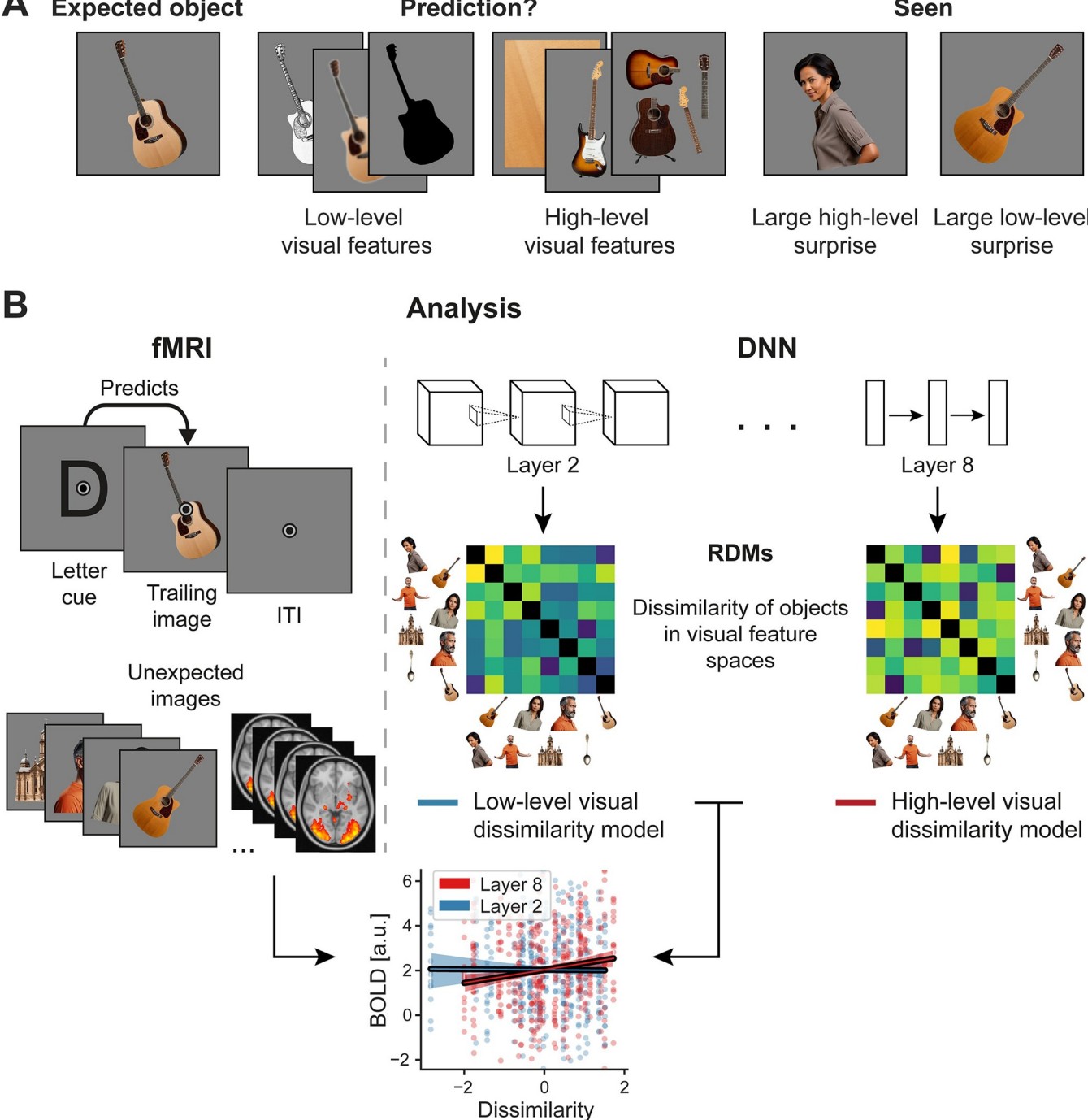

**Fig 4. Analytical approach. (A)** If you expect to see the first guitar on the left, what kind of visual features does visual cortex predict? Low-level visual features, illustrated next to the expected image, concern local oriented edges, spatial frequency, and similar properties. High-level visual features entail more complex visual representations, such as texture-like features [32], core object parts and their relationships, features commonly shared between instances of an object, irrespective of the specific depiction. Depending on which features are predicted, the 2 "seen" images will result in different prediction error magnitudes. The image of the woman is very different in high-level visual features, but shares some local orientation with the expected guitar, hence resulting primarily in high-level visual surprise. On the other hand, the image of the other guitar is very different in terms of low-level visual features, as it is differently rotated compared to the expected guitar, but it is still a guitar and thus shares high-level visual features. The key question of the analysis is whether and where in the visual system low-level or high-level surprise results in larger prediction errors. **(B)** Analysis procedure. The left side shows a single trial with a letter cue predicting a specific image. Below multiple unexpected images are illustrated, which were presented on other trials. The right side depicts the extraction of low-level and high-level visual feature dissimilarity from layer 2 (low-level) and layer 8 (high-level) of the visual DNN. Dissimilarity of the unexpected seen image relative to the expected stimulus was then added as a parametric modulator in the first level GLMs of the fMRI analysis. For illustration purposes, the graph at the bottom

uses data from one example participant to depict how BOLD responses (example data from V1; ordinate) are modulated as a function of low-level (blue) and high-level (red) visual dissimilarity (abscissa). A positive slope thus indicates that the more dissimilar a seen image was relative to the expected stimulus the more vigorous the neural response. Dots indicating individual dissimilarities (i.e., distances of seen unexpected images compared to the expected image) are arranged in rows, reflecting the granularity of the available surprise distances for this example participant. In the example data, the image of the unexpected woman would hence result in larger prediction errors compared to the image of the unexpected guitar, because of the larger high-level surprise. Critically, this analysis only uses BOLD data from unexpected image trials. Additional control variables for task relevance (animacy) and word meaning, discussed in more detail later, were also included. We performed this parametric modulation analysis in a voxel-wise fashion across the whole brain. DNN, deep neural network; GLM, general linear model.

the analysis rationale. If low-level visual features, such as local oriented edges and spatial frequency, are predicted in a specific cortical area (e.g., V1) then prediction error magnitudes should scale with low-level surprise in that area. As an example, expecting a specific image of a guitar but seeing an image of another guitar from a different angle should yield a large prediction error as the 2 images are different from a low-level visual feature standpoint. On the other hand, if high-level visual features, such as more abstract and general guitar features (e.g., a neck and guitar body) are predicted, invariant to local orientation, then the unexpected guitar should not yield a strong prediction error, whereas seeing an image of an unexpected category should result in a large prediction error (even if low-level features are similar). Based on our analyses of DNN alignment with fMRI localizer data and prior work [27], we decided to use layer 2 of the visual DNN as low-level feature model and layer 8 (before softmax of the output layer) as high-level visual feature models. This allowed for maximal differentiation from the low-level model. To model how the BOLD response changed as a function of how visually surprising an unexpected stimulus was in terms of low-level (layer 2) and high-level (layer 8) visual features respectively, we used the dissimilarity of each surprising image, compared to the stimulus expected on that trial, as parametric modulators in the fMRI GLM analysis (for details, see Materials and methods: Data analysis). Using this approach, variance not uniquely associated with either regressor will not contribute to the observed results. Thus, in addition, we included multiple control variables, such as task-relevant stimulus animacy and word-level (semantic) dissimilarity.

Results, depicted in Fig 5A, demonstrated that surprise responses scaled significantly with high-level visual dissimilarity (layer 8) in visual cortex, encompassing early and intermediate visual areas (cluster size 779 voxel, 6,232 mm$^3$; S1 Table contains additional details). That is, the more an unexpected stimulus diverged from the expected image in terms of high-level visual features, the more the sensory response increased in magnitude. Surprisingly, we did not find any modulation of neural responses by low-level visual dissimilarity (layer 2) anywhere in visual cortex. In other words, even in EVC prediction error magnitudes were modulated by high-level but not low-level visual surprise, as indexed by layer 2. In the example in Fig 4A, this corresponds to the unexpected image of the woman eliciting a larger prediction error in V1 compared to the unexpected guitar. On the other hand, the low-level surprise elicited by the unexpected guitar would not result in an additional up-regulation of prediction errors in visual cortex beyond the associated high-level visual surprise.

Our whole-brain results were corroborated by a region of interest (ROI) analysis, depicted in Fig 5B (ROI masks are illustrated in Fig 3B). Results showed a strong difference between high- and low-level visual feature models in modulating BOLD surprise responses (main effect of model: $F_{(1,32)} = 42.78$, $p < 0.001$, $\eta_p^2 = 0.57$). We found reliable modulations of surprise responses by high-level visual, but no significant modulation by low-level visual features, in primary visual cortex (V1: Layer 8: $t_{[32]} = 6.79$, $p < 0.001$, $d_z = 1.18$; Layer 2: $W = 190$, $p = 0.159$, $d_z = -0.32$, $BF_{10} = 0.58$), intermediate visual areas in the lateral occipital complex (LOC: Layer 8: $W = 126$, $p = 0.017$, $d_z = 0.55$; Layer 2: $t_{(32)} = -0.68$, $p = 0.504$, $d_z = -0.12$, $BF_{10}$

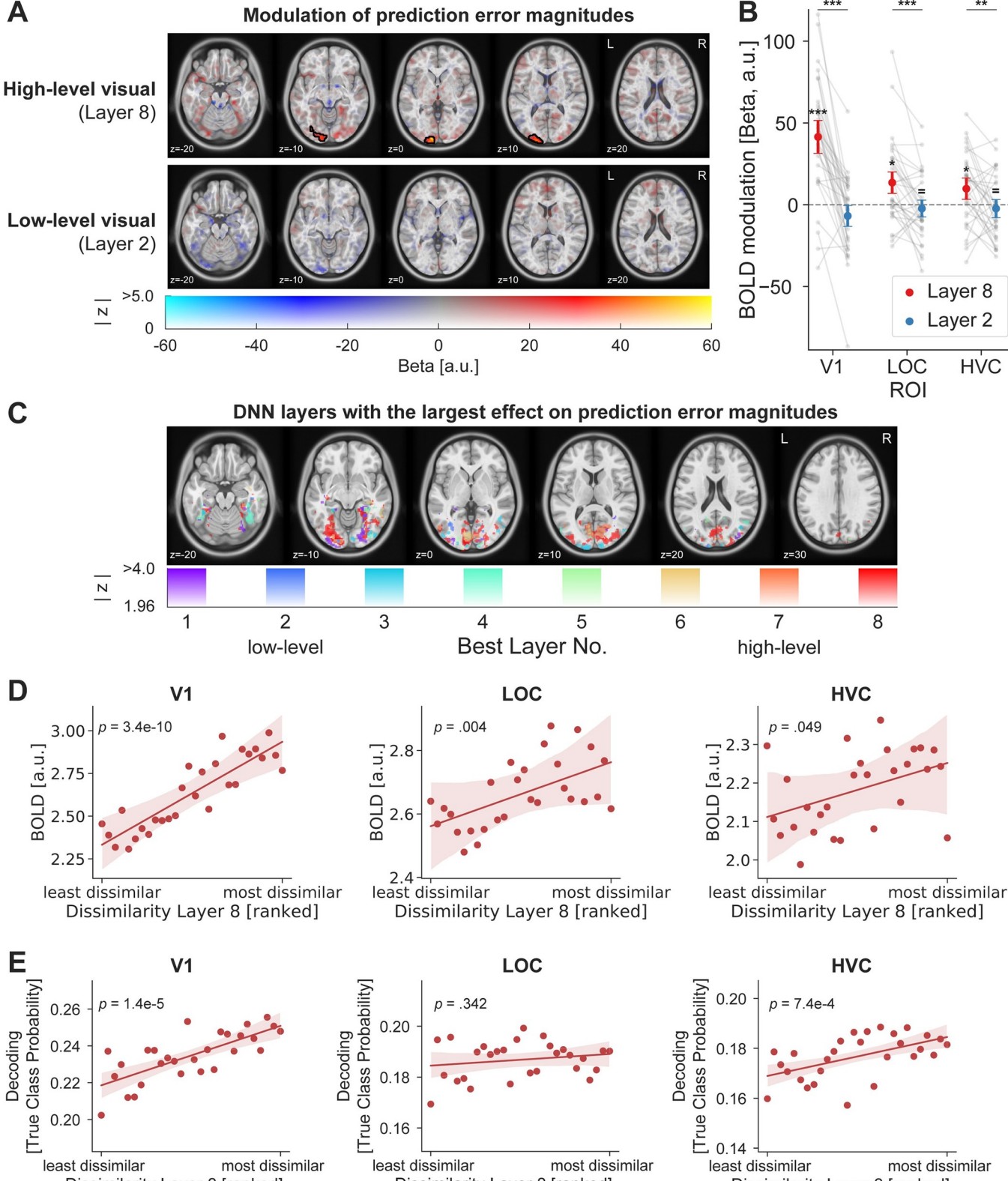

**Fig 5. Prediction error magnitude scales with high-level visual feature surprise. (A)** Whole-brain results assessing the modulation of surprise responses as a function of high-level (top row) and low-level (bottom row) visual feature dissimilarity. The top row shows that surprise responses to unexpected images were increased if the image was more distant from the expected image in terms of high-level visual features. Color indicates the beta parameter estimate of the

parametric modulation, with red and yellow representing increased responses. Black outlines denote statistically significant clusters (GRF cluster corrected). No significant modulation of sensory responses was observed by low-level visual surprise. **(B)** ROI analysis zooming in on ROIs in early visual (V1), intermediate (LOC), and HVC (encompassing occipito-temporal sulcus and fusiform cortex). Results mirror those of the whole-brain analysis, with significant modulations of the visual responses by high-level visual surprise (red), but not low-level visual surprise (blue). Error bars indicate the 95% within-subject confidence intervals. Gray dots denote individual subjects. *P* values are FDR corrected. *** $p < 0.001$, ** $p < 0.01$, * $p < 0.05$, $^=$ $BF_{10} < 1/3$. **(C)** Prediction errors preferentially scale with high-level visual features (layer 8 and layer 7) throughout most of the visual system, including EVC, LOC and HVC. Color indicates the DNN layer with the largest effect (explained variance) on scaling the neural responses to surprising inputs. Cold colors (purple–blue) represent early layers (i.e., low-level visual features), while warm colors (yellow–red) indicate late layers (i.e., high-level visual features). Analysis was masked to visual cortex and thresholded at a liberal $z \geq 1.96$ (i.e., $p < 0.05$, two-sided) to explore the landscape of prediction error modulations across DNN layers. Results strongly contrast with those observed for prediction-free visual responses during the localizer (Fig 3A). **(D, E)** ROI analysis regressing BOLD responses (**D**) or decoded true class probability (**E**) onto high-level visual dissimilarity. Results indicate a monotonic relationship between high-level surprise and BOLD responses across all 3 ROIs, as well as decoding performance in V1 and HVC. The chance level for decoding the true class probability is 0.125. For display purposes dissimilarities were ranked and averaged across participants, while regression models were fit per participant on the correlation distances. Data and code that support these findings are available at: https://doi.org/10.34973/8e49-2012. DNN, deep neural network; EVC, early visual cortex; HVC, higher visual cortex; LOC, lateral occipital complex; ROI, region of interest; V1, primary visual cortex.

= 0.23) and high-level visual cortex (HVC: Layer 8: $t_{(32)} = 2.59$, $p = 0.029$, $d_z = 0.45$; Layer 2: $t_{(32)} = -0.70$, $p = 0.586$, $d_z = -0.12$, $BF_{10} = 0.23$). Contrasting the modulation by layer 8 against layer 2 dissimilarity confirmed that layer 8 modulated visual surprise responses significantly more than layer 2 in V1 ($t_{(32)} = 8.05$, $p < 0.001$, $d_z = 1.40$), LOC ($t_{(32)} = 4.24$, $p < 0.001$, $d_z = 0.74$), and HVC ($t_{(32)} = 2.85$, $p = 0.008$, $d_z = 0.50$). Hence, the larger the visual dissimilarity of a surprising stimulus in terms of high-level visual features, the more vigorous the visual response across the ventral visual stream, including V1. No corresponding modulation by low-level visual features was observed, suggesting that high-level features are predominantly reflected in visual prediction error signals. A complementary control analysis (S2 Fig), using stimulus uninformative voxels (i.e., chance or near chance-level decoding accuracy) showed no modulation by either layer 2 or layer 8 dissimilarity, suggesting that the modulation of visual responses by layer 8 dissimilarity is specific to stimulus selective voxels.

In a subsequent analysis, we asked which DNN layer explained most neural variance of the prediction error response. We analyzed how neural responses to unexpected stimuli were scaled as a function of surprise indexed by each layer of the DNN. To this end, we regressed layers 1 to 8 dissimilarity onto single trial parameter estimates and determined for each voxel which layer had the largest explained variance. Results (Fig 5C) showed that prediction error magnitudes primarily scaled with high-level visual surprise (layer 8 and layer 7) across most parts of the ventral visual stream, including EVC, LOC, and HVC. We note additional minor clusters in HVC scaled by intermediate layer 4, as well as layers 1 and 3 in EVC and LOC, suggesting that some neural populations scaled with intermediate and low-level surprise as well. In sum, these results present a stark contrast to the modulation of responses during the prediction-free localizer (Fig 3A) where a clear gradient from early-to-late layers was observed, with early layers dominating EVC representations. In contrast, prediction errors appear to be scaled preferentially by high-level visual surprise across the visual system, including EVC.

Next, we assessed the shape of the response modulation by high-level visual surprise by regressing BOLD responses for unexpected stimuli onto layer 8 dissimilarity using an ROI approach. Results, shown in Fig 5D, show that the increased BOLD response to surprising stimuli follows a positive monotonic association across the dissimilarity spectrum in all 3 ROIs (V1: $t_{(32)} = 9.35$, $p < 0.001$, $d_z = 1.63$, mean $r = 0.09$; LOC: $t_{(32)} = 3.27$, $p = 0.004$, $d_z = 0.57$, mean $r = 0.03$; HVC: $t_{(32)} = 2.05$, $p = 0.049$, $d_z = 0.36$, mean $r = 0.02$).

In addition to modulating BOLD responses, it is possible that high-level surprise may also result in sharper visual representations. To test this, we performed a decoding analysis, trained on the independent localizer data and tested on the main task data. Decoding accuracy of the object images (for details, see: Materials and methods: Statistical Analysis: Decoding as

function of surprise) increased with layer 8 dissimilarity in V1 ($t_{(32)}$ = 5.50, $p < 0.001$, $d_z$ = 0.96, mean $r$ = 0.08) and HVC ($t_{(32)}$ = 3.89, $p < 0.001$, $d_z$ = 0.68, mean $r$ = 0.05), but not LOC ($t_{(32)}$ = 0.96, $p$ = 0.342, $d_z$ = 0.17, mean $r$ = 0.07). In other words, the more surprising an unexpected stimulus was in terms of high-level features, the better it could be decoded from V1 and HVC. To summarize, visual responses and the fidelity of the visual representations appeared to monotonically increase with high-level visual surprise across major parts of the ventral visual system.

## Prediction error scaling with high-level visual surprise is not explained by task-relevance, semantic surprise, or inherent properties of the DNN architecture

Multiple alternative explanations could account for a correlation of prediction error magnitudes with high-level visual features. To rule out alternative accounts for our observations, we included multiple control variables as parametric modulators in our GLM analysis. First, layer 8 representations from an untrained (i.e., random) but otherwise identical DNN were included. This contrast ruled out that the inherent structure of the DNN architecture or correlations in the input images caused the scaling of prediction errors with late layer representations. Second, animacy category was included in the GLM to assess whether high-level visual modulations result from a correlation of high-level visual features with the task-relevant dimension of animacy. Since participants had to distinguish between animate and inanimate entities in the images, prediction error modulations could potentially reflect task responses. Finally, because high-level visual features may significantly correlate with semantic, word category level surprise, we also contrasted the high-level visual model against a word2vec [33,34] derived model aimed at indexing nonvisual, semantic surprise.

These controls revealed that high-level visual surprise (layer 8) best accounted for the data, significantly outperforming an untrained random layer 8 model, animacy category, word category (semantic), and the low-level visual surprise models in explaining prediction error magnitudes (Fig 6A). Statistically significant clusters were found in EVC as well as intermediate visual areas in LOC, and HVC in some contrasts. The exact extent of the modulation varies slightly between contrasts, but overall corroborate that prediction error magnitudes mainly result from high-level visual feature surprise and that none of the control variables likely account for the observed results. Corresponding whole-brain figures contrasting the control parametric modulators against baseline (no modulation) can be found in S3 Fig. Additionally, variance inflation factors (VIFs; S2 Table) were smaller for layer 2 (VIF = 1.42) than layer 8 surprise (VIF = 1.88), suggesting that the absence of a significant modulation by low-level surprise was not due to problems with variance partitioning due to collinearity of the predictors in the GLM. In fact, all VIFs were significantly lower than the suggested threshold of VIF < 5 [35].

An ROI analysis (Fig 6B) of the same 4 contrasts confirmed the whole-brain results. We observed reliable differences between models, which differed across ROIs (main effect of model: $F_{(4,128)}$ = 12.20, $p < 0.001$, $\eta_p^2$ = 0.28; interaction ROI by model: $F_{(4.40,140.93)}$ = 11.09, $p < 0.001$, $\eta_p^2$ = 0.26). Specifically, we found significantly stronger modulations of BOLD responses by high-level visual dissimilarity compared to all other 4 parametric modulators in V1 (paired $t$ tests: all FDR corrected $p < 0.001$; all $d > 0.99$; see S3 and S4 Tables for details). Similar, albeit less pronounced results were found in LOC (Layer 8 versus Layer 2: $p$ = 0.001, $d_z$ = 0.74; Layer 8 versus Animacy category: $p$ = 0.010, $d_z$ = 0.61; Layer 8 versus Word2Vec: $p$ = 0.060, $d_z$ = 0.47; Layer 8 versus Random layer 8: $p$ = 0.078, $d_z$ = 0.44) and HVC (Layer 8 versus Layer 2: $p$ = 0.028, $d_z$ = 0.50; Layer 8 versus Animacy category: $p$ = 0.081, $d_z$ = 0.40;

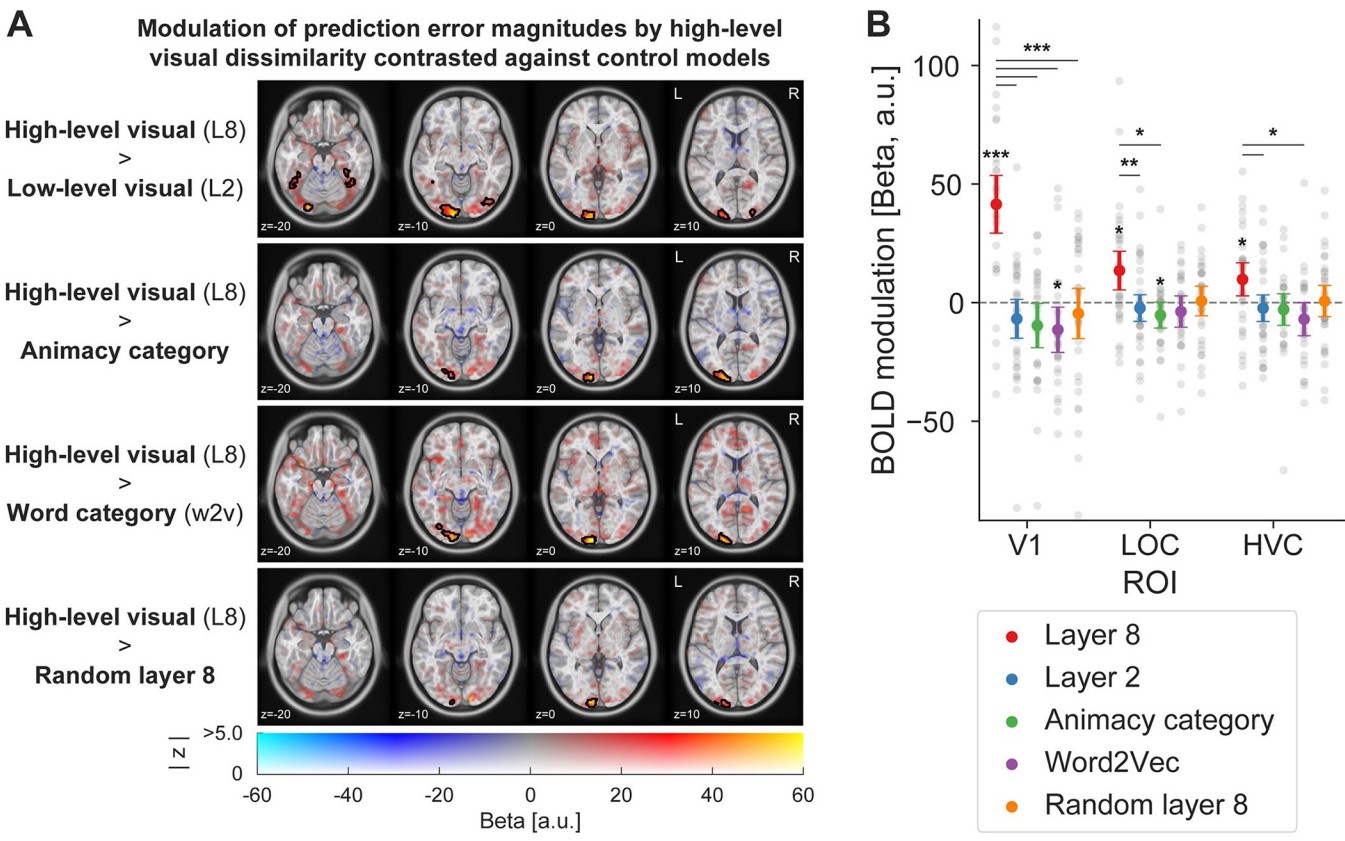

**Fig 6. Prediction error magnitudes are best explained by high-level visual feature dissimilarity. (A)** Whole-brain contrasts of the high-level visual feature model (layer 8) contrasted against 4 control variables. The top row shows that high-level visual models performed significantly better than low-level visual models (layer 2). Similarly, high-level visual surprise better accounted for prediction error magnitudes than the task-relevant animacy category of the unexpected stimuli (second row) and the semantic, word category surprise model (word2vec; third row). The bottom row shows that high-level visual dissimilarity significantly better explained prediction error magnitudes compared to an untrained but otherwise identical DNN layer 8. **(B)** ROI analysis including primary (V1), intermediate (LOC), and high-level visual cortex (HVC). Results confirm the whole-brain results, showing significant modulations of BOLD responses by high-level visual surprise (red) compared to low-level visual (blue), response category (green), and word category surprise (purple). Error bars indicate the 95% within-subject confidence intervals. Gray dots denote individual subjects. *P* values are FDR corrected. *** $p < 0.001$, ** $p < 0.01$, * $p < 0.05$. Data and code that support these findings are available at: https://doi.org/10.34973/8e49-2012. DNN, deep neural network; HVC, higher visual cortex; LOC, lateral occipital complex; ROI, region of interest; V1, primary visual cortex.

Layer 8 versus Word2Vec: $p = 0.016$, $d_z = 0.54$; Layer 8 versus Random layer 8: $p = 0.189$, $d_z = 0.32$). An additional negative modulation of prediction error magnitudes by word category surprise was observed in V1 ($p = 0.020$, $d_z = -0.60$), and by animacy category surprise in LOC ($p = 0.039$, $d_z = -0.51$), suggesting that prediction errors may be attenuated for more semantically dissimilar surprising images in V1 and for stimuli of a difference animacy category in LOC compared to the expected image. Finally, to ensure that our results were not dependent on the exact ROI mask definition and mask sizes, we repeated the analysis for an alternative ROI mask definition (using all stimulus driven voxels within the anatomically defined masks) and across multiple mask sizes. Results were qualitatively identical across ROI definitions (S4 Fig) and mask sizes (S5 Fig).

## Discussion

Hierarchical predictive processing theories [1–4,36] have received significant attention as they propose a fundamental framework for cortical computation. Numerous studies have

corroborated the main tenets of predictive processing, such as demonstrating that sensory responses to surprising inputs are enhanced compared to expected ones, likely reflecting larger sensory prediction errors [9,15–17,37]. However, while evidence for this core mechanism of predictive processing has been shown across modalities, paradigms, and species [12,14,38], it remains unknown what kind of surprise is reflected in these putative prediction errors. Here, we set out to elucidate the nature of the neural modulation by visual prediction errors and thus what information is predicted across the visual hierarchy.

## Visual prediction errors scale with high-level visual surprise

Using fMRI and representational distance measures derived from a visual DNN, our data showed that throughout multiple visual cortical areas sensory responses to unexpected images scale with the representational distance of a seen unexpected stimulus relative to the expected input. Specifically, responses monotonically scaled with high-level visual feature surprise: the larger the high-level deviation from expectation, the larger the prediction error response. Interestingly, and in contrast to feedforward processing, even early visual areas, such as V1, predominantly responded to high-level visual over low-level surprise, while being best explained in terms of low-level features in cases where no prediction was possible (localizer). These results are in line with a recent study demonstrating that firing rates in macaque V1 correlate with spatial predictability of high-level and not low-level visual features [39]. The increased neural activity for high-level visual surprise in EVC, a region that is not tuned to high-level visual features during feedforward processing, may suggest that predictions are relayed top-down and hence result in the observed inheritance of feature surprise from higher areas in earlier processing stages. Thus, our results support and extend previous studies by demonstrating that (1) top-down inheritance during predictive vision generalizes across species and recording modalities; and (2) crucially appears to be a general principle of visual sensory processing evident across multiple cortical areas that together encompass the ventral visual stream [25,40]. Additionally, the more surprising objects were in terms of high-level features, the better they could be decoded by a linear classifier. Therefore, our data suggested that both the magnitude but also the stimulus information contained in the visual response scaled with high-level surprise. This may imply a possible functional role in enhancing visual representations of inputs that do not match our high-level predictions of the world. Moreover, our results further demonstrate that predictive signatures and top-down inheritance of high-level feature surprise can arise, at least in humans, with little exposure to the predictive regularities, requiring only several dozen exposures rather than extensive exposure as in the case of studies in nonhuman primates [25]. This flexibility of the visual system to learn and rapidly utilize novel sensory priors to generate high-level predictions to inform sensory processing in earlier stages of the hierarchy further supports the hypothesis that predictive top-down signaling is a ubiquitous and general principle of visual processing.

What kind of mechanism may underlie the here observed modulations of sensory responses? Earlier neurophysiological work in macaques have found that the visual prediction error response in IT has roughly the same latency as the response to the stimulus itself [15], whereas high-level surprise affects a late stage of neural responses [25,39]. We speculate that our observed fMRI BOLD correlates reflect a similar late-stage top-down modulation of sensory processing. This interpretation aligns well with our observation that the feedforward response, mostly reflected during the prediction-free localizer, is dominated by local tuning properties (e.g., low-level features in V1), while recurrent processing due to prediction during the main task, relying on feedback and reflecting high-level visual surprise, takes time to arise given the necessary computations and signal relaying across multiple cortical areas. Taken

together, our results suggest that following the activation of high-level areas, during the initial feedforward sweep, a prediction is subsequently relayed down the processing hierarchy modulating the sustained phase of neural responses across earlier areas.

Elaborating further on this account, one possible explanation for the observed results is that predictions allow the visual system to settle on a valid perceptual interpretation more quickly, as predictions match the bottom-up inputs. In contrast, unexpected input may require stronger [15] and more prolonged processing within the visual system to arrive at the best interpretation of the current inputs, potentially leading to the larger BOLD signal and superior decoding of stimulus representations observed here. Put differently, our results may suggest that the larger the high-level difference between what is expected and what is observed the more cycles of recurrent processing along the ventral visual stream may be necessary to solve object recognition. To clarify, here we are not suggesting that visual processing in V1 primarily reflects high-level visual information, but rather that its modulation as a function of expectation reflects high-level surprise, potentially implemented by cortico-cortical feedback.

Our analyses also demonstrated that the modulation of prediction error responses by high-level visual surprise was not explained by the task-relevant animacy category dimension or by word-level (semantic) surprise. These results thus suggest that visual prediction errors are predominantly modulated by high-level visual and not abstract linguistic or response-related surprise. This preference aligns with our earlier proposition of facilitated perceptual inference, with visual prediction errors scaling primarily with visual surprise due to the role of top-down feedback in constraining visual interpretations in lower areas. Consequently, because high-level visual areas in the ventral stream encode primarily high-level visual features, this surprise signal is expressed in terms of high-level visual features instead of abstract non-visual representations.

In the whole-brain analyses (Figs 5A and 6A), we found reliable up-regulations of prediction error amplitudes by high-level visual surprise in the visual system, but not in the motor system or other areas outside the visual system, such as inferior frontal gyrus or anterior insula, known to generate prediction errors especially when predictions are task-relevant [10,41–43]. This suggests that the modulation by high-level visual surprise primarily concerns facilitated perceptual inference rather than facilitated decision-making or response initiation. Yet, this does not mean that predictions do not facilitate these processes as well. The focus of our investigation was the modulation of neural responses for different unexpected inputs. A direct comparison of unexpected compared to expected inputs (i.e., expectation suppression) can be found in S6 Fig and matches previous studies showing additional prediction error signatures in decision, attention, and motor processing-related areas [10,41–43], suggesting that prediction facilitates processing across multiple cortical systems. A supplemental analysis decoding stimulus identity is reported in S5 Table.

## Flexible prediction (error) tuning

While we found no reliable up-regulation of prediction error magnitudes by any of the control models (also see: S3 Fig), this does not imply that the visual system exclusively encodes high-level visual surprise. Although we dismissed that animacy category explained our results, this does not rule out that task requirements shaped the acquisition and generation of predictions, and consequently the scaling of prediction errors. Indeed, it is possible that the focus on high-level features, due to the task requirements, shaped what kind of features were predicted, given the substantial effect that tasks can have on visual processing [44]. Thus, our results are also consistent with recent models of adaptive efficient coding [45]. Specifically, because our task required a high-level decision, such task demands might be reflected in the adaptive

compression (silencing) of non-task relevant sensory representations due to top-down feedback, thereby resulting in the observed scaling of early visual responses with high-level visual surprise. Moreover, evolutionary it is advantageous to develop flexible predictions, allowing the visual system to adapt to environmental requirements. In line with this hypothesis, recent evidence suggests that semantic (word level) priors can be used to generate category specific sensory predictions, even in early visual cortex [46]. Therefore, more abstract, semantic representations could modulate visual prediction errors under certain conditions. Additionally, considering the nature of the utilized DNN, it is probable that our high-level visual feature model captured a wide range of category variability, with animacy representing just one such (binary) dimension.

On the other hand, it is also plausible that visual prediction errors can reflect low-level features, particularly if required by the environment. Indeed, in the present data small additional clusters across multiple areas scaled with intermediate-level surprise, suggesting variety in the surprise reflected in visual prediction errors. Furthermore, employing a different task specifically targeting low-level features, or using stimuli that are predictable predominantly in terms of these features [9] might indeed lead to strong representations of low-level surprise. However, here we observe for naturalistic stimuli, which inherently support a hierarchy of predictions, that lower-level predictions may be subsumed by high-level ones, mirroring recent results in language processing [47]. In addition, it is likely that there are limits to the flexibility imposed by the architecture and representational constraints of the visual system. As argued above, high-level visual predictions may be relayed top-down and constrain visual interpretations in lower areas, because neurons in higher visual areas are tuned to high-level features. In contrast, it is less likely that high-level visual areas have the necessary representational structure and acuity to predict detailed low-level features, and hence these areas may be unable to constrain visual interpretations in EVC for low-level visual features to the same degree as for high-level predictions. Nonetheless, we believe that it is essential for future work to chart the extent to which prediction error tuning is flexibly adjusted to reflect environmental and task demands.

## Limitations

There are some limitations to the present research. A superior low-level visual feature model may have yielded prediction error modulations by such features. In addition, the specific stimulus set could have discouraged or obscured low-level surprise modulations. Nevertheless, we demonstrated that early DNN layers best explain feedforward visual activity in EVC during the prediction-free localizer runs. This observation underscores that the low-level feature model employed in our study reliably accounted for the visual responses elicited by our stimuli in the EVC, outperforming intermediate or high-level feature models. Consequently, the absence of low-level surprise modulations in our results cannot be attributed to a general inadequacy of the low-level model. Moreover, we chose a commonly used visual DNN. This and similar models have repeatedly been shown to share representational geometry with visual cortex [27–31] despite important differences between how DNNs and cortex represent stimuli (e.g., receptive fields). Finally, there is no evidence for a strong positive, but sub-threshold, modulation of visual responses by low-level visual surprise evident in either the whole-brain or ROI results. This suggests that it is unlikely that a quantitative issue, such as low statistical power due to a subpar low-level feature model or non-ideal stimulus set for evoking low-level visual predictions, explains the current results. A limitation of the present approach is that neighboring layers of visual DNNs tend to correlate and hence explain significant shared variance of visual representations (S8 Fig), thus contraindicating the inclusion of all layers into the

same GLM. For this reason, our key analyses focused on 2 layers, layer 2 as low-level and layer 8 as high-level visual feature model. Future work could improve on the generalization of the present results by utilizing different, ideally more complex stimulus sets and improved feature models, thereby further investigating whether (other aspects of) low-level features may modulate visual prediction errors.

The word category (semantic) model has similar limitations. While word2vec and related models have been widely used to index semantic dissimilarity [48–50], they may no longer constitute state-of-the-art models. Thus, given a superior semantic feature model [51] or different task, visual prediction errors could be modulated by semantic surprise. However, while improvements in the semantic model are possible, it seems unlikely that incremental improvements in the model can fully account for the observed results, especially in EVC, given that semantic modulations tend to arise later in the processing hierarchy. In sum, while we cannot rule out that in some circumstances other models, stimuli, or tasks could result in low-level visual or linguistic semantic features modulating sensory prediction errors, our data indicates a propensity of the visual system to scale prediction errors in terms of high-level visual surprise.

The precise features represented by visual DNNs are poorly understood. Visualizing features (S1 Fig) can help to gain an intuition for the feature tuning, but ultimately hand-crafted and better controlled feature models might be necessary to further elucidate the precise features that are reflected in visual prediction errors. That said, our conclusions are not dependent on layers 2 and 8 specifically; see S7 Fig for comparable results using layers 1 or 3 instead of layer 2 as low-level, and layer 7 instead of layer 8 as high-level visual surprise model. In other words, we do not propose that layer 8 of the utilized DNN is particularly important, rather it should be seen as one instance of a high-level feature space correlating with visual representations and hence scaling with visual cortical surprise.

Interestingly, surprise modulations appeared to be more pronounced in V1 compared to LOC or HVC. This raises the question of whether such a distinction reflects a functionally significant difference between these ROIs, such as V1 acting as a high-resolution blackboard [52] particularly sensitive to predictive modulations, or if this result reflects a more mundane consequence of differences in neurovascular coupling that leads to superior signal-to-noise ratios in V1. The exact nature of this observation remains to be elucidated.

## Adaptation and attention-based accounts

One concern in predictive processing studies is differentiating prediction from stimulus repetition effects such as neural adaptation. Stimulus adaptation is not a viable explanation for the present results, because all stimuli were shown equally often. Moreover, our results concern modulations of prediction error responses to unexpected stimuli based on how different they were from the expected stimuli, thereby further ruling out repetition frequency or adaptation as explanations.

Another possible interpretation is that surprising stimuli capture attention or increase general arousal, which subsequently amplifies neural responses [53]. While we cannot conclusively rule out this alternative account, there are multiple factors suggesting that it is not the primary factor. First, attention accounts [53] emphasize that generic prediction error-like responses may arise due to the more demanding processing associated with responding to unexpected compared to expected inputs. While constituting a viable account of previous studies, it is critical to emphasize that the results reported here exclusively analyzed differential responses to unexpected inputs, and thus are not affected by differences in processing demands associated with responding to expected compared to unexpected inputs. Second, an arousal or attention-

based account must explain why arousal or attention allocation would scale primarily with high-level visual surprise, rather than semantic features or the task-relevant dimension of animacy. Moreover, even if we treat the observed modulation as an effect of attention or arousal, high-level visual surprise must be detected before attention reallocation, raising the question where this surprise detection occurs if not reflected by the modulation reported here. Therefore, the predictive processing account detailed earlier appears to provide a more parsimonious explanation. However, previous work showed that predictions are gated by attention [10], thus suggesting that both attention and prediction are necessary for the here observed modulations. In line with this interpretation, this prior study also suggests that both stimulus specific and unspecific modulations contribute to prediction errors, with the latter likely reflecting generic changes in arousal or attention [10]. On either account, it is likely that the present modulations in EVC reflect a consequence of the downstream surprise detection in later (visual) areas.

Moreover, merging prediction and attention accounts, we can derive another interpretation. Following prediction error computation in higher (visual) areas, a surprise signal could be relayed down the visual system. This surprise signal, instead of constituting a prediction, may be a scalar value reflecting the magnitude of the high-level surprise (or on a predictive coding account, the precision of the high-level prediction error) and result in a gain modulation of sensory neurons in earlier areas. This surprise signal could serve to allocate attention, facilitate learning, as well as update synaptic weights and thus promote encoding of statistical contingencies following highly surprising events. Sources of this modulatory feedback could be higher visual areas that compute high-level prediction errors or the perirhinal cortex, thereby closely linking high-level surprise with novelty detection and facilitating learning [54].

## Conclusions

We find that visual prediction errors, a key feature of hierarchical predictive processing theories, primarily reflect high-level visual surprise, including in early visual cortex. These results show a striking dissociation between the bottom-up tuning profile in early visual cortex and its modulation by high-level surprise. They suggest that visual surprise is computed at higher levels of the visual ventral stream and surprise signals are subsequently broadcast to earlier areas in the cortical hierarchy. Collectively, our results thereby bolster a central mechanism of hierarchical predictive processing—the reliance of perceptual inference on prediction and prediction error generation, thus reinforcing the crucial role of predictions in perception.

## Materials and methods

### Participants and data exclusion

In total 40 healthy, right-handed participants were recruited from the Radboud University research participation system. Of these, data from 2 participants were incomplete due to the participants withdrawing from the experiment. In addition, we excluded data from 2 further participants due to poor MRI data quality, caused by excessive motion during MRI scanning in one case and anterior coil failure in the other. Furthermore, data of 3 participants were excluded because of subpar behavioral performance (see Data analysis, Data exclusion). Thus, in total data from 33 participants (21 female, age 23.8 ± 4.5, mean ± SD) were included in the final sample.

The study followed institutional guidelines, the Declaration of Helsinki, and was approved by the local ethics committee (CMO Arnhem-Nijmegen, now METC Oost-Nederland) under the blanket approval "Imaging Human Cognition" (2014/288) granted to the Donders Centre

for Cognitive Neuroimaging, Radboud University Nijmegen. Written informed consent was obtained before study participation and participants were compensated 10€/hour.

## Stimuli and experimental paradigm

During the experiment, participants were exposed to pairs of letter cues and full-color images of various categories while recording fMRI. On each trial the letter probabilistically predicted the identity of the image. The expected image was 7 times more likely to follow its associated letter cue compared to each unexpected image.

**Stimuli.** Full-color images were selected from a database of 233 photographs, originally collected for a previous study [46]. All image stimuli are shared together with the experiment code; see: Materials and methods: Software and data availability. The image database included multiple exemplar images of various categories, including animate (dogs, dolphins, elephants, feet, hands, women and men, male and female faces, swans, tigers) and inanimate objects (cars, airplanes, churches, guitars, hammers, houses, spoons). Of the original 233 stimuli in the database, 20 stimuli were excluded as outliers. Specifically, we used hierarchical agglomerative clustering to cluster image representations of layers 2 and 8 of AlexNet trained on ecoset (for more details, see: Deep Neural Network). We then excluded any outliers, such as same category objects in a different cluster than all other exemplars of the same category in terms of layer 8 representations. From the remaining 213 stimuli, 8 images were selected for each participant, including 4 animate and 4 inanimate images. The selection of the 8 images was optimized using the following criteria. First, we derived RDM from the layers of interest (layer 2 and layer 8) from the DNN using correlation distance. We then randomly selected 8 images (4 animate and 4 inanimate) and calculated the variances within layer RDMs. Additionally, we calculated the across layer RDM correlation. Finally, we maximized the within layer variances and minimized the across layer correlation. This procedure is a simple method to select image samples that maximized the detectability of effects within RDMs, while also minimizing the correlation between RDMs, thereby increasing our ability to detect distinct contributions of the RDMs from the 2 layers of interest. For each participant, we selected a set of 8 images using this procedure. Images were randomly paired with a specific letter for each participant. Thus, most participants saw different images and in the case of the same image being used for more than 1 participant, the individual letter-image pairings were different as well. We made this choice to rule out, by design, that specific images or cue-image pairs drive the results, for example due to high representational similarity for some cue-image pairs. Moreover, because participants were shown different stimuli (8 selected from a set of 213 stimuli), our results are unlikely to be contingent on any specific set of 8 stimuli. Similarly, because the pairing of the letter cues and image stimuli was randomized between participants, even participants who saw the same stimuli did not learn the same letter-image pairs (approximately $3.7 \times 10^{21}$ possible combinations), thereby effectively ruling out that specific cue-image combinations account for the results.

Images were presented in the center of the screen, subtending a maximum of $6 \times 6$ degrees of visual angle. The exact image size depended on the shape of the specific object. A fixation bulls-eye, outer circle 0.5 degrees of visual angle, was displayed on top of the center of the image.

**Experimental paradigm and procedure.** On each trial participants were presented with a letter cue for 500 ms, followed by an image for 500 ms without interstimulus interval. The letter cues were predictive of the image with approximately 50% reliability—the TPM is depicted in Fig 1B. Thus, participants could predict the identity of the images given the letter. Participants were not informed about these regularities. Instead, they were tasked to categorize the

entities in the images as animate or inanimate as quickly and accurately as possible. Thus, while learning the statistical regularities was not required to perform the task, the regularities could be used to facilitate task performance. To promote statistical learning, participants were required to withhold the response if the letter cue was a vowel (a, e, i, o, u), thereby directing attention also towards the letter cues. No-go letters (vowels) were not associated with any specific stimulus and appeared in addition to the regularities depicted in Fig 1B. No-go trials were discarded from all analyses. Responses on each trial were given by button press (right index or middle finger) as soon as the image appeared, with a maximum allowed reaction time of 1,500 ms before a trial would be considered a miss. Trials were separated by an intertrial interval of on average 5,000 ms (range 3,000 ms to 12,000 ms, sampled from a truncated exponential distribution), displaying only the fixation bulls-eye.

Each run, that is one continuous fMRI data acquisition, consisted of 128 trials (approximately 13 min). During a run, the TPM shown in Fig 1B was presented once. Thus, each unexpected cue stimulus pair was shown exactly one time (8 × 7 combinations = 56 unexpected trials) and each expected pair was shown 7 times (8 expected pairs × 7 repetitions = 56 expected trials), with the remaining 16 trials being no-go trials. Trial order was randomized, except for excluding repetitions of the same cue stimulus pair on 2 consecutive trials. Participants performed 4 runs per session and 2 sessions, resulting in a total of 8 fMRI runs. In addition to the fMRI runs, the experiment also included an additional 2 behavioral blocks in each session.

The behavioral blocks were identical to the fMRI runs, except for a shorter intertrial interval (average 2,500 ms, range 1,500 ms to 7,500 ms) and an adjusted TPM. Specifically, expected pairs were shown 3 times more often during behavioral blocks compared to fMRI runs (i.e., each pair had 21 repetitions instead of 7) to facilitate statistical learning. Moreover, twice as many no-go trials were shown to compensate for the increased number of expected trials. Thus, behavioral blocks consisted of a total of 256 trials per block.

On day 1, participants first performed one functional localizer run (see: Functional localizer), followed by a short practice of the main task using different letters and images. Then, the 4 fMRI main task runs followed, and finally 2 behavioral blocks were performed outside the MRI. On the next day, the order of runs was reversed. Thus, participants first did the 2 behavioral blocks, then the 4 fMRI main task runs, followed by another run of the functional localizer, and finally an anatomical scan was acquired.

**Functional localizer.** A functional localizer, depicted in Fig 1C, was performed to define object selective LOC, constrain anatomical ROI masks using independent fMRI data, and to perform RSA to validate the RDMs derived from the DNN layers of interest. The functional localizer used a block design, presenting one stimulus at a time for 12,000 ms, flashing every 800 ms (500 ms on, 300 ms off), during each miniblock. Miniblock order was randomized, thus precluding prediction of the next stimulus, but excluded direct repetitions of the same stimulus. Participants were tasked to press a button whenever the image changed in brightness. The image noticeably increased in brightness (approximately 200%) at a random cycle exactly once per miniblock, except for during the first 3 and last 2 cycles. Each image was presented during 4 miniblocks per localizer run. In addition, a phase scrambled version of each image was shown, with each scrambled image being repeated in 2 miniblocks. As during the main fMRI task runs, images subtended 6 × 6 degrees of visual angle and a fixation bulls-eye was displayed at the center of the image throughout the entire run.

## fMRI data acquisition

MRI data was acquired on a Siemens 3T Prisma and a 3T PrismaFit scanner, using a 32-channel head coil. Functional images were acquired using a whole-brain T2*-weighted multiband-

6 sequence (TR/TE = 1,000/34 ms, 66 slices, voxel size 2 mm isotropic, FOV = 210 mm, 60˚ flip angle, A/P phase encoding direction, bandwidth = 2,090 Hz/Px). Anatomical images were acquired using a T1-weighted MP-RAGE sequence (GRAPPA acceleration factor = 2, TR/TE = 2,300/3.03 ms, voxel size 1 mm isotropic, 8˚ flip angle).

## Statistical analysis

**Behavioral data analysis.** Behavioral data was analyzed in terms of reaction time (RT) and accuracy. Trials with too fast (<100 ms) or too slow (>1,500 ms) RTs were excluded. Only trials with correct responses were analyzed for the RT analysis. RTs and accuracy were calculated for expected and unexpected image trials separately. Additionally, unexpected trials were split into trials requiring the same button press as the expected image (unexpected same) and trials requiring a different response (unexpected different). RT and accuracy data were analyzed using one-way repeated measures ANOVAs with 3 levels. Following significant results, post hoc $t$ tests were conducted, and $p$ values corrected using the Holm–Bonferroni method. Behavioral data was also used to reject outliers based on poor overall response accuracy and speed (for details see: Data exclusion).

**Effect size and error calculation.** We provide the following estimates of effect size to support statistical inference. For $t$ tests, we report Cohen's $d_z$ [55], for Wilcoxon signed-rank tests matched-pairs rank-biserial correlation ($r$), and partial eta-squared ($\eta_p^2$) for repeated measures ANOVAs.

Within-subject confidence intervals, as depicted in Figs 5 and 6, are calculated using the within-subject normalization procedure introduced by Cousineau [56] with Morey [57] bias correction.

## fMRI data preprocessing

**MRI preprocessing pipeline description generated by fMRIprep.** MRI data was preprocessed using *fMRIPrep* 22.1.0 [58], which is based on *Nipype* 1.8.5 [59].

**Anatomical data preprocessing.** The T1-weighted (T1w) image was corrected for intensity non-uniformity (INU) with `N4BiasFieldCorrection` [60], distributed with ANTs 2.3.3 [61], and used as T1w-reference throughout the workflow. The T1w-reference was then skull-stripped with a *Nipype* implementation of the `antsBrainExtraction.sh` workflow (from ANTs), using OASIS30ANTs as target template. Brain tissue segmentation of cerebrospinal fluid (CSF), white matter (WM) and gray matter (GM) was performed on the brain-extracted T1w using `fast` (FSL 6.0.5.1; [62] RRID:SCR_002823). Brain surfaces were reconstructed using `recon-all` (FreeSurfer 7.2.0; [63]; RRID:SCR_001847), and the brain mask estimated previously was refined with a custom variation of the method to reconcile ANTs-derived and FreeSurfer-derived segmentations of the cortical GM of Mindboggle [64] (RRID: SCR_002438). Volume-based spatial normalization to one standard space (MNI152NLin2009-cAsym) was performed through nonlinear registration with `antsRegistration` (ANTs 2.3.3), using brain-extracted versions of both T1w reference and the T1w template. The following template was selected for spatial normalization: *ICBM 152 Nonlinear Asymmetrical template version 2009c* [65] RRID:SCR_008796; TemplateFlow ID: *MNI152NLin2009cAsym*).

**Functional data preprocessing.** For each of the 10 BOLD runs per subject (across all tasks and sessions), the following preprocessing was performed. First, a reference volume and its skull-stripped version were generated by aligning and averaging 1 single-band references (SBRefs). Head-motion parameters with respect to the BOLD reference (transformation matrices, and 6 corresponding rotation and translation parameters) are estimated before any spatio-temporal filtering using `mcflirt` (FSL 6.0.5.1) [66]. BOLD runs were slice-time corrected to

0.445s (0.5 of slice acquisition range 0 s to 0.89 s) using `3dTshift` from AFNI ([67]; RRID: SCR_005927). The BOLD time series (including slice-timing correction when applied) were resampled onto their original, native space by applying the transforms to correct for head-motion. These resampled BOLD time-series will be referred to as preprocessed BOLD in original space, or just preprocessed BOLD. The BOLD reference was then co-registered to the T1w reference using `bbregister` (FreeSurfer) which implements boundary-based registration [68]. Co-registration was configured with 6 degrees of freedom. First, a reference volume and its skull-stripped version were generated using a custom methodology of *fMRIPrep*. Several confounding time series were calculated based on the preprocessed BOLD: framewise displacement (FD), DVARS, and 3 region-wise global signals. FD was computed using 2 formulations following Power (absolute sum of relative motions, Power and colleagues [69]) and Jenkinson (relative root mean square displacement between affines, Jenkinson and colleagues [66]). FD and DVARS are calculated for each functional run, both using their implementations in *Nipype* (following the definitions by Power and colleagues [69]). The 3 global signals are extracted within the CSF, the WM, and the whole-brain masks. Additionally, a set of physiological regressors were extracted to allow for component-based noise correction (*CompCor*, [70]). Principal components are estimated after high-pass filtering the preprocessed BOLD time series (using a discrete cosine filter with 128 s cutoff) for the 2 *CompCor* variants: temporal (tCompCor) and anatomical (aCompCor). tCompCor components are then calculated from the top 2% variable voxels within the brain mask. For aCompCor, 3 probabilistic masks (CSF, WM, and combined CSF+WM) are generated in anatomical space. The implementation differs from that of [70] in that instead of eroding the masks by 2 pixels on BOLD space, a mask of pixels that likely contain a volume fraction of GM is subtracted from the aCompCor masks. This mask is obtained by dilating a GM mask extracted from the FreeSurfer's *aseg* segmentation, and it ensures components are not extracted from voxels containing a minimal fraction of GM. Finally, these masks are resampled into BOLD space and binarized by thresholding at 0.99 (as in the original implementation). Components are also calculated separately within the WM and CSF masks. For each CompCor decomposition, the *k* components with the largest singular values are retained, such that the retained components' time series are sufficient to explain 50 percent of variance across the nuisance mask (CSF, WM, combined, or temporal). The remaining components are dropped from consideration. The head-motion estimates calculated in the correction step were also placed within the corresponding confounds file. The confound time series derived from head motion estimates and global signals were expanded with the inclusion of temporal derivatives and quadratic terms for each [71]. Frames that exceeded a threshold of 0.5 mm FD or 1.5 standardized DVARS were annotated as motion outliers. Additional nuisance time series are calculated by means of principal components analysis of the signal found within a thin band (crown) of voxels around the edge of the brain, as proposed by Patriat and colleagues [72]. The BOLD time series were resampled into standard space, generating a preprocessed BOLD run in MNI152NLin2009cAsym space. First, a reference volume and its skull-stripped version were generated using a custom methodology of *fMRIPrep*. All resamplings can be performed with a single interpolation step by composing all the pertinent transformations (i.e., head-motion transform matrices, susceptibility distortion correction when available, and co-registrations to anatomical and output spaces). Gridded (volumetric) resamplings were performed using `antsApplyTransforms` (ANTs), configured with Lanczos interpolation to minimize the smoothing effects of other kernels. Non-gridded (surface) resamplings were performed using `mri_vol2surf` (FreeSurfer).

For more details of the pipeline, see the section corresponding to workflows in fMRIPrep's documentation (https://fmriprep.org/en/latest/workflows.html).

**Additional preprocessing.** After preprocessing using fMRIprep, additional fMRI data preprocessing steps were performed using FSL FEAT and Nilearn, including high-pass filtering (128 s cutoff) and spatial smoothing (5 mm fwhm).

## fMRI data analysis

Univariate fMRI analyses consisted of fitting voxel-wise general linear models (GLMs) to each participant's run data, using an event-related approach. Stimuli were modeled as events of 500 ms duration with the onset corresponding to the onset of the stimuli. Hence, cues, presented 500 ms before stimulus onset, were not explicitly modeled. Events were convolved with a double gamma hemodynamic response function. Expected and unexpected image trials were modeled as separate regressors. Moreover, parametric modulators were added to the design matrix, reflecting how different an unexpected image was compared to the expected image. Specifically, a parametric modulator was added based on the representational dissimilarity of the unexpected compared to the expected image on a given trial in terms of layer 2 and layer 8 representations of AlexNet (see: Deep neural network data for details). Importantly, while the distance metric (surprise) thus depends on the distance of the seen surprising image from the expected (unseen) image on each trial, the parametric modulation only concerns trials with unexpected images. Thereby this analysis does not contrast fMRI BOLD responses to unexpected compared to expected inputs, but exclusively assesses the scaling of unexpected images by surprise. Additional parametric modulators were included to serve as control variables, consisting of animacy category, word category (*word2vec*) and layer 8 distance from an untrained (random) AlexNet instance. The parametric modulators were z scored before being added to the design matrix. A regressor of no interest was included for no-go trials. First order temporal derivatives of these regressors were also added to the GLM.

Nuisance regressors were added, consisting of 6 standard motion parameters (rotation and translation in x, y, and z), FD, CSF, and WM. All nuisance regressors were derived from fMRIprep. To deal with temporal autocorrelation FSL's FILM with local autocorrelation correction was used [73]. Parameter estimates were averaged across runs using a fixed effects analysis and across participants using FSL FEAT's mixed-effects model. All fMRI analyses were performed in normalized space (*MNI152NLin2009cAsym*).

Contrasts of interest were the modulation of the BOLD response by the parametric modulators indexing the representational dissimilarity of the unexpected compared to the expected image, especially, AlexNet layer 2 and layer 8. Statistical maps were corrected for multiple comparisons using Gaussian random-field cluster thresholding, as implemented in FSL FEAT 6.0, with a cluster formation threshold of $z \geq 3.29$ (i.e., $p < 0.001$, two-sided) and a cluster threshold of $p < 0.05$.

**Regression of BOLD onto dissimilarity.** In addition to the parametric modulation analysis, we performed a complementary analysis, regressing single trial BOLD parameter estimates (see: Single trial parameter estimation) onto the z scored dissimilarity metrics. This regression was performed for each participant separately, in a whole-brain and ROI fashion. The resulting slope coefficients for each voxel or ROI, indexing the modulation of BOLD responses as a function of dissimilarity were then subjected to a one-sample $t$ test contrasting the obtained slope against zero (no modulation). For the whole-brain analysis additional spatial smoothing of 3 mm fwhm was applied (i.e., total smoothing 8 mm). Finally, we colored each voxel according to which layer had the largest effect on the visual responses, indexed by explained variance. We thresholded this whole-brain analysis to a liberal $z \geq 1.96$ (i.e., $p < 0.05$, two-sided) to explore the landscape of the predictive modulations.

**Single trial parameter estimation.** Single trial parameter estimates were obtained using a least squares separate approach [74,75]. A GLM was fit per trial to the BOLD data using Nilearn, where each design matrix contained a regressor for the trial of interest (iterating over all trials) and regressors of no interest for the remaining images, split by image identity, as well as a regressor for no-go trials. Nuisance regressors were also included, consisting of 6 motion parameters (rotation and translation in x, y, and z), FD, CSF, and WM. From this, we extracted the parameter estimates for each trial, with particular interest in the parameter estimates of unexpected appearances of the stimuli.

**Region of interest (ROI) analysis.** ROIs were defined a priori, based on previous studies [10], and consisted of early visual cortex (V1), intermediate object-selective areas in the LOC and higher visual cortex (HVC) consisting of primarily of temporal occipital fusiform cortex. These 3 ROIs constitute well studied early, intermediate, and late ventral visual stream areas. ROI masks were defined both anatomically and functionally for each participant. First, we used Freesurfer (http://surfer.nmr.mgh.harvard.edu/, RRID:SCR_001847) for cortex segmentation and parcellation [63,76], run as part of the fMRIprep pipeline. The resulting V1 labels were transformed to native volumetric space using `mri_label2vol`. Additional atlas annotations were extracted from the Destrieux Atlas [77]. The LOC mask was formed by combining 2 Freesurfer labels in the lateral occipital cortex (middle occipital gyrus (lateral occipital gyrus) and inferior occipital gyrus and sulcus). The HVC mask was obtained by merging 3 labels corresponding to higher ventral visual stream areas (lateral occipito-temporal gyrus (fusiform gyrus), lateral occipito-temporal sulcus, and medial occipito-temporal sulcus (collateral sulcus) and lingual sulcus). Left and right hemisphere masks were combined into bilateral masks and dilated using a 3 mm Gaussian kernel. Overlapping voxels between the 3 ROI masks (V1, LOC, and HVC) were assigned to the mask containing less voxels. We then resampled the masks to standard space (MNI152NLin2009cAsym) for each participant.

Object-selective LOC. In line with previous studies [10,78,79], and to ensure that LOC contained stimulus selective neural populations, we further constrained the anatomical LOC masks per participant using data from an independent localizer run. In brief, we contrasted the response to intact compared to phase scrambled versions of the images, thereby obtaining voxels that respond more intact stimuli for each participant.

Voxel selection. In line with previous studies [17], we selected within each ROI the 200 voxels most informative about the depicted stimulus. To this end, we performed a decoding analysis on the localizer data and selected the voxels affording best stimulus decoding (see: Decoding searchlight analysis). To ensure that our results generalize beyond the a priori selected, but arbitrary ROI size, we repeated all ROI analyses with masks ranging from 100 to 500 voxels (800 to 4,000 mm$^3$). Additionally, we repeated the ROI analyses using a more liberal ROI mask definition, using all stimulus driven voxels (localizer data) within the anatomically defined masks.

ROI analysis. For each subject and ROI, we extracted the contrast parameter estimates from the second level GLMs (subject level, averaged across runs) of the parametric modulators of the main task using the ROI masks refined using the localizer data. These parametric modulators were then tested against zero (i.e., no modulation) using one-sample t tests or Wilcoxon signed-rank tests as appropriate, as well as contrasted against one another. Resulting p values were FDR corrected for the number of ROIs and tested parametric modulators.

## Deep neural network

We used AlexNet, trained on ecoset [27], to derive representational dissimilarity estimates of the utilized stimuli. Early and late layers of this network have previously been shown to map

well onto early and late visual cortex representations, respectively [27]. We extracted representational dissimilarities using correlation distance (1 –correlation) from all layers. RDMs of 10 different instances of the DNN were averaged to minimize effects specific to any particular instance of the DNN [80]. We were especially interested in layer 2 representing an early low-level visual feature space, and layer 8 constituting a high-level visual feature space. For more information on the DNN, see: [27]. For each participant, we used the DNN derived RDMs to index how different each unexpected stimulus was compared to the expected stimulus in terms of low-level feature dissimilarity (layer 2) and high-level feature dissimilarity (layer 8). The image dissimilarity scores were z scored before being included as parametric modulators in the first level GLMs (see: fMRI data analysis).

Untrained network control analysis. In addition, we included RDMs from layer 8 of the same DNN, using the same procedure, but used an untrained (i.e., random) network instance. RDMs from this untrained network serve as a control condition to rule out DNN architecture or stimulus set specific contributions to the results.

### Word-level and animacy category feature spaces

Additionally, we derived RDMs using word embeddings to approximate a semantic, nonvisual dissimilarity metric. Specifically, we used word2vec [33,34], pretrained on the Google News corpus (word2vec-google-news-300), to derive the pairwise dissimilarity between category words describing our image stimuli (airplane, car, church, dog, elephant, face, foot, guitar, hammer, hand, house, man, spoon, swan, tiger, woman).

Animacy category. As animacy was a task-relevant feature, we also derived an RDM indexing animacy. This RDM thus constitutes an object category dissimilarity and indirectly also task-response metric. To create this parametric modulator, we created a vector with zeros for unexpected stimuli with the same animacy category as the expected stimulus, and hence also response, and ones representing unexpected stimuli with a different animacy category. Dissimilarities from these control metrics were included in the first level fMRI GLM.

### Representational similarity analysis

We validated the DNN derived RDMs using independent localizer data. Specifically, we used RSA [22] to test whether the utilized DNN layer RDMs did significantly resemble the visual cortical RDMs. To this end, we extracted parameter estimates for each stimulus compared to baseline from the first level GLMs of the localizer runs for each participant using a searchlight approach (6 mm radius). We then z scored these parameter estimates per voxel and computed the representational dissimilarity in each searchlight sphere between the different stimuli, as indexed by the cosine distance between the vectors of the parameter estimates. This resulted in the neural RDM. We then correlated the lower triangular of the neural RDMs with the lower triangular of the DNN derived RDMs using Kendall's Tau. For each voxel, we selected the DNN layer that explained most neural variance. Finally, we z transformed the resulting correlation coefficients and subjected them to one-sample $t$ tests across subjects for each voxel. A significant test would thus indicate that the neural RDMs and DNN-derived RDMs shared representational geometry, suggesting that the DNN RDM was useful in explaining neural variance during the localizer run. We considered this a requirement for proceeding with the main task analysis of prediction error representations.

### Decoding searchlight analysis

An additional searchlight (radius 6 mm) was used to decode stimulus identity across the whole brain, using linear support vector machines (SVMs). We first derived for each participant

separately, single trial parameter estimate maps from the localizer run using the least squares separate procedure outlined before. On these single trial parameter estimates, using the searchlight approach, the SVM was trained and tested using 4-fold cross-validation. The labels supplied to this decoding analysis were the image identities. Thus, the resulting decoding maps indicated the ability to decode stimulus identity during the localizer runs. This decoding map was subsequently used to refine the ROI masks (see: Region of interest (ROI) analysis).

### Decoding as function of high-level surprise

In addition to the univariate analysis, we regressed decoding performance onto high-level (layer 8) surprise. Specifically, for each ROI and participant we trained a multi-class SVM on the single trial parameter estimates from the prediction-free localizer data. Next, we predicted class membership probabilities on a per-trial basis during the main task using single trial parameter estimates of all unexpected image trials. Finally, per participant, we regressed the true class probabilities derived from the decoder onto high-level (layer 8) surprise. Across participants we performed inference by submitting the obtained beta coefficients to a one-sample $t$ test. A significant positive association would thus suggest that decoding performance increases with layer 8 dissimilarity.

### Bayesian analyses

We evaluated any nonsignificant frequentist tests shown in Fig 5 using equivalent Bayesian analyses to assess evidence for the absence of an effect of the low-level visual feature parametric modulator. To this end, JASP 0.17.1.0 [81] with default settings was used for Bayesian $t$ tests with a Cauchy prior width of 0.707. Qualitative interpretations of the resulting Bayes factors were based on Lee and Wagenmakers [82].

### Data exclusion

Data were excluded from analysis based on 2 independent criteria. First, we excluded participants due to low-quality fMRI data, quantified in terms of high mean FD, percentage of FD exceeding 0.2 mm (FD%), high temporal derivative of variance over voxels (DVARS) and low temporal signal to noise ratio (tSNR). These 4 image quality metrics were derived using MRIQC [83] for each run. For details concerning the calculation of each image quality metric, see Esteban and colleagues [83]. Subsequently, we averaged these metrics across runs within participants and compared each participant to the sample mean. Participants with any image quality metric worse than the sample mean plus (or minus depending on the metric) 2 SD was rejected from further analysis. Two participants were excluded due to these fMRI image quality metrics.

In addition, we excluded participants based on subpar behavioral performance, indicating a lack of attention or task compliance. Using a similar approach as for the MRI quality metrics, we rejected each participant with an average behavioral response accuracy or reaction time 2 SD worse than the sample mean. Three participants were excluded for poor behavioral performance.

### Software and data availability

Python 3.7.4 (Python Software Foundation, RRID:SCR_008394) was used for data processing, analysis and visualization using the following libraries: NumPy 1.18.1 (RRID:SCR_008633) [84,85], Pandas 1.1.4 [86], NiLearn 0.9.1 (RRID:SCR_001362), Scikit-learn 0.24.2 [87], SciPy 1.5.3 (RRID:SCR_008058) [88], Matplotlib 3.1.3 (RRID:SCR_008624) [89], Gensim (using

word2vec, RRID:SCR_014776) [90], and Seaborn 0.11.2 (RRID:SCR_018132) [91]. A conda environment yml file is included with the code. MRI data was preprocessed using fMRIprep [58] and analyzed using FSL 6.0 (FMRIB Software Library; Oxford, UK; www.fmrib.ox.ac.uk/fsl; RRID:SCR_002823) [73]. Whole-brain results were visualized using Slice Display [92], a MATLAB (2022b; The MathWorks, Natick, Massachusetts, United States, RRID:SCR_001622) data visualization toolbox. DNN feature visualization was performed using Lucid, adjusted for Ecoset trained AlexNet (https://github.com/KietzmannLab/lucid-kietzmannlab). Stimuli were presented with Presentation software (version 20.2, Neurobehavioral Systems, Berkeley, California, RRID:SCR_002521). Some images depicted in Fig 4 were generated using SDXL [93]. All data, stimuli, RDMs, as well as experiment and analysis code required to replicate the results reported in this paper are available from the Radboud Data Repository: https://doi.org/10.34973/8e49-2012.

## Supporting information

**S1 Fig. DNN feature representations. (A)** Feature visualization using maximal activation. In brief, starting from random noise, an image was optimized to maximally activate a particular channel of a layer. The optimization function was the negative of the spatial activation of that layer and channel (for the code and more details, see: https://github.com/KietzmannLab/lucid-kietzmannlab). From the resulting images, maximally driving the channel responses of the layer of interest, we can thus qualitatively interpret the kinds of features represented in that layer. Depicted are example images maximally driving activation of 3 channels (y axis) of layers 2, 5, and 8 (x axis). Images on the left are characterized by repeating clear orientations, thus suggesting that layer 2 represents low-level, Gabor-like features. Layer 5, in the center column, contains larger scale orientation features, but also textures. On the right side, image exemplars maximally driving layer 8, reflect more high-level visual features, such as abstract, irregular patterns and textures. Thus, the feature visualization results suggest 2 primary conclusions. One, layer 2 does reflect typical low-level visual features, such as orientation. Second, layer 8 reflects higher level visual features compared to previous layers, most notably complex patterns and irregular textures, with little low-level visual feature representation (e.g., no evidence of orientation tuning). Therefore, the feature visualization results support our assumption that layer 2 largely reflects low-level and layer 8 high-level visual features. **(B)** Decoding of object image category and Gabor orientation from DNN layer activations. If the qualitative interpretation above is correct, we would expect that layer 2 contains significantly more low-level visual feature information compared to layer 8. In contrast, we would expect that more category information is available in layer 8 representations. We tested these hypotheses by performing 2 decoding analyses. First, we extracted the DNN activations of layer 2 and 8 in response to the 213 object stimuli used in the experiment. Next, we trained and tested a multiclass linear SVM, using 8-fold cross validation, to classify the 17 categories of these stimuli, thereby assessing how much category information is available to be read out by a linear classifier in each layer's activations. Second, we performed the same classification analysis using layer 2 and 8 activation maps in response to Gabor patches. Here, classes were defined as Gabor orientations, spanning 0 to 169 degrees orientation, split evenly across 17 classes (orientations in steps of approximately 10.5 degrees). Exemplars within each class had different spatial frequencies (6 to 18 cycles), approximately matching the number of exemplars within each class in the object image dataset. Decoding accuracy, using 8-fold cross validation, is depicted on the ordinate for layers 2 and 8 (abscissa). Object category decoding (green), was significantly better in layer 8 compared to layer 2. In contrast, Gabor orientation (orange) could be decoded better in layer 2 than 8. Moreover, there was a strong interaction, with Gabor

orientation being decoded substantially better than image category in layer 2, and vice versa in layer 8. In sum, these results further support that layer 2, in contrast to layer 8, preferentially represents low-level visual feature information, while layer 8 is more tuned towards high-level information, such as object categories. Data and code that support these findings are available at: https://doi.org/10.34973/8e49-2012.
(TIF)

**S2 Fig. No modulation of neural responses by visual surprise in stimulus uninformative voxels.** Control ROI analysis using stimulus uninformative voxels in early visual (V1), intermediate (LOC), and higher visual cortex (HVC; encompassing occipito-temporal sulcus and fusiform cortex). ROI masks were defined using the same procedure as outlined in the region of interest (ROI) analysis paragraph of the Materials and methods sections, except for that voxel selection was performed by choosing the least informative (i.e., lowest decoding performance) voxels for object identity decoding during the localizer runs. No modulation of neural responses by high-level (layer 8) or low-level (layer 2) surprise was observed in any ROI containing voxels uninformative about the stimuli. These results further suggest that the modulation of visual responses by high-level surprise (Fig 5) is specific to stimulus-selective voxels and not an unspecific global surprise signal. Error bars indicate the 95% within-subject confidence intervals. Gray dots denote individual subjects. $^=$ $BF_{10} < 1/3$. Data and code that support these findings are available at: https://doi.org/10.34973/8e49-2012.
(TIF)

**S3 Fig. Whole-brain control analysis.** Whole-brain results assessing the modulation of surprise responses as a function of feature dissimilarity indexed by animacy category (top row), word-level semantic surprise (middle row), and a random (i.e., untrained) but otherwise identical visual DNN instance (bottom row). Results show no reliable modulation by any of these control models anywhere in cortex, except for a small negative modulation by word category surprise (word2vec) in precuneus cortex, outside of stimulus-driven voxels. Thus, unexpected stimuli of a similar semantic category as the expected stimulus may elicit larger BOLD responses. This modulation could reflect an increased requirement for processing resources to distinguish different exemplars of the same category, albeit its small size and localization to voxels in superior parts of precuneus cortex, that were not stimulus driven during the localizer run, makes an interpretation challenging. Color indicates the beta parameter estimate of the parametric modulation, with red and yellow representing increased responses. Black outlines denote statistically significant clusters (GRF cluster corrected). Data and code that support these findings are available at: https://doi.org/10.34973/8e49-2012.
(TIF)

**S4 Fig. Visual responses scale with high-level surprise across the visual hierarchy in stimulus-driven voxels. (A)** Control ROI analysis using all stimulus driven within the anatomically defined V1, LOC, and HVC masks. For ROI mask creation, stimulus driven was defined as a significant activation ($z > 3.1$) of voxels by the presentation of the object images during the independent localizer run. Thus, compared to the ROI masks used in the main analyses, these ROI masks are larger, encompassing all visually driven voxels within the ROIs. Average mask size in voxels: V1 = 1,254, LOC = 2,443, HVC = 1,383. Results show that prediction error magnitudes are best explained by high-level visual feature dissimilarity across all 3 ROIs. Error bars indicate the 95% within-subject confidence intervals. Gray dots denote individual subjects. $P$ values are FDR corrected. *** $p < 0.001$, ** $p < 0.01$. **(B)** Voxels included in the ROI masks using all stimulus-driven voxels within the anatomically defined masks. Color indicates the ROI: Blue = V1, Purple = LOC, Orange = HVC. Notably, the resulting masks are

significantly larger and extend further anterior in HVC compared to the original decoding-based masks. Opacity indicates the proportion of participants whose individual masks included the voxel. For visualization, full opacity corresponds to a proportion of 0.1, with voxel inclusion proportions reaching up to 1. Data and code that support these findings are available at: https://doi.org/10.34973/8e49-2012.
(TIF)

**S5 Fig. Control ROI analysis across different ROI mask sizes.** Modulation of surprise as a function of high-level (layer 8; first panel), low-level (layer 2; second panel), animacy category (third panel), word level (word2vec; fourth panel), and random model (last panel) surprise. Reliable modulations of prediction error magnitudes were found for high-level visual surprise across all tested ROI sizes (100–500 voxel) in V1, as well as most mask sizes in LOC (100–250 and 350–450 voxel) and TOFC (150–350 voxel). No statistically significant modulation was found for low-level visual surprise or the random layer 8 model for any ROI or mask size. A negative modulation of BOLD responses was found for word level (word2vec) surprise in V1 for several ROI masks (150–300 and 400–450 voxel) and 3 small mask sizes in LOC (100–200 voxel) for animacy category. Thus, overall results closely match the results reported in Fig 6 for most mask sizes, confirming that the observed results are largely robust to variations in ROI size. Error bars depict 95% confidence intervals. Data points with black outline indicate statistical significance at $p < 0.05$ (FDR corrected for the number of ROIs and models) compared to zero (i.e., no modulation). Data and code that support these findings are available at: https://doi.org/10.34973/8e49-2012.
(TIF)

**S6 Fig. Expectation suppression.** Generic prediction errors (unexpected–expected) were calculated from the voxel-wise GLM including regressors for expected and unexpected appearances of the image stimuli, as well as the parametric modulators. Here, we depict the contrast unexpected–expected, thus indexing differences in neural responses contingent on whether the stimulus was expected or unexpected. Color indicates the beta parameter estimate, with red and yellow representing increased responses to unexpected stimuli. Black outlines denote statistically significant clusters (GRF cluster corrected). Significant clusters can be seen in visual cortex, particularly in temporal occipital fusiform cortex, anterior insula, and inferior frontal gyrus. Additional significant cluster, not visible here, were found in superior parietal lobule, paracingulate gyrus, and supplementary motor cortex; see S1 Table for details. These areas closely match previous reports of prediction error responses [10,42]. The data that support these findings are available at: https://doi.org/10.34973/8e49-2012.
(TIF)

**S7 Fig. Control analyses using neighboring layers of the primary prediction error analysis.** To ensure that our results were not specific to the DNN layers used in the primary analyses (see the Results section for details on the layer selection), we repeated the ROI analyses using surprise as indexed by the direct neighbors of our layers of interest. **(A)** ROI analysis in V1, LOC, and HVC using DNN layer 1 (blue) and layer 7 (red) as low-level and high-level surprise, respectively. **(B)** ROI analysis using layer 3 (blue) and layer 7 (red) surprise. Layer 9 was not explored, as it constitutes the class output layer of the DNN after softmax, and thus is categorical in nature. While compared to the primary analyses (Fig 6), using layer 2 and layer 8 surprise, quantitative differences are evident, qualitatively the results remain identical. That is, high-level surprise (here layer 7), best explains prediction error magnitudes compared to other surprise metric, particularly in V1. In contrast to layer 8 surprise, modulations by layer 7 were not quite as pronounced in later visual areas. Like layer 2, both layers 1 and 3 surprise failed to

account for modulations of visual prediction errors. Error bars indicate the 95% within-subject confidence intervals. Gray dots denote individual subjects. *P* values are FDR corrected. ***
*p* < 0.001, ** *p* < 0.01, * *p* < 0.05. Data and code that support these findings are available at: https://doi.org/10.34973/8e49-2012.
(TIF)

**S8 Fig. Representational similarity analyses show that visual responses during the prediction-free localizer were explained by a gradient of low-level to high-level visual features going up the ventral visual hierarchy.** Early visual cortex (EVC) responses align more closely with early and intermediate DNN layers, indicative of low-level visual feature processing. Higher visual cortex (HVC) areas, like the fusiform gyrus, show a greater correlation with late DNN layers, representing high-level visual feature processing. Red to yellow cluster indicate statistically significant correlations between the layer RDMs (layers 1–8 separately for each panel), thresholded at z > 3.1 (i.e., *p* < 0.001, uncorrected). ROI masks are illustrated as colored outlines with black = V1, green = LOC, and blue = HVC. A strong contrast between earlier convolutional layers (1–5) and later dense layers (6–8) is evident. Particularly early layers strongly map onto EVC, while later layers map onto late ventral visual areas. We also note that while our ROI definition does separate the ventral visual stream roughly into 3 separate stages corresponding to early, intermediate, and high-level visual cortex, our HVC definition does not include major anterior HVC clusters particularly correlating with late DNN layer representations. Critically, our results do not depend on the specific ROI mask definition depicted, because larger ROI masks than those depicted here (see S5 Fig) and an alternative ROI mask definition using all stimulus-driven voxels (see S4 Fig) resulted in highly similar results as those obtained using the above depicted masks. Data and code that support these findings are available at: https://doi.org/10.34973/8e49-2012.
(TIF)

**S1 Table. Brain areas showing significant modulations of BOLD responses (GRF cluster corrected).** Listed are the contrasts of the parametric modulators (layer 8, layer 2, word category, animacy category, and random layer 8), as well as the contrast "unexpected minus expected stimuli" (Expectation suppression; S6 Fig) with corresponding area labels, numbers of voxels in the cluster, *p* value of the cluster, and peak z statistic. MNI coordinates indicate the X, Y, Z coordinates of the center of gravity for the cluster, as derived by FSL FEAT's `cluster` function, in MNI space. Area labels are based on the center of gravity for the cluster and, especially for large clusters, additional areas encompassed by the cluster.
(PDF)

**S2 Table. Variance inflation factor (VIF) for each regressor.** VIFs were computed on the first-level (i.e., run-level) fMRI design matrices, as supplied to FSL FEAT, and subsequently averaged across runs and participants. Overall, obtained VIFs are low—commonly VIFs >5 are considered problematic. Additionally, the VIF for the low-level visual surprise regressor is lower (i.e., better) compared to high-level visual surprise, suggesting that the absence of a modulation of neural responses by low-level surprise is unlikely to be caused by problems with variance partitioning due to collinearity of the predictors in the GLM.
(PDF)

**S3 Table. Results of one sample *t* tests and Wilcoxon signed rank test contrasting parameter estimates of the parametric modulators against zero (no modulation).** We found reliable modulations by the high-level visual model (layer 8) throughout all 3 ROIs, encompassing early (V1) and high level (HVC). *P* values are FDR corrected.
(PDF)

**S4 Table. Results of paired *t* tests and Wilcoxon signed rank test, contrasting the parameter estimates of the parametric modulators in a pair-wise fashion within each ROI.** The high-level visual model (layer 8) modulated sensory responses significantly more than any of the 4 control models in V1, as well as some models in LOC and HVC. *P* values are FDR corrected.
(PDF)

**S5 Table. Decoding of stimulus identity across expectation conditions.** For each participant and within each ROI, we extracted the single-trial parameter estimates (BOLD) from the localizer runs. On this data, we trained an 8-class classifier, with the 8 classes corresponding to the 8 objects seen by the participant. Next, we tested the classifier on the single-trial parameter estimates from the main task runs. To improve SNR, we averaged the single-trial estimates for each object and expectation condition (expected or unexpected) within runs. Finally, we averaged the obtained decoding accuracies across runs to obtain a mean decoding accuracy for expected and unexpected trials per participant. On average decoding of the object stimuli was well above chance (12.5%), but did not reliably differ between expected and unexpected stimuli in any of the 3 ROIs (V1: $t_{(32)} = -0.51$, $p = 0.613$; LOC: $t_{(32)} = -1.02$, $p = 0.315$; HVC: $t_{(32)} = 0.55$, $p = 0.588$).
(PDF)

## Acknowledgments

We thank Maartje J. Graauwmans for assistance with data acquisition, Daniel Anthes with DNN programming, and Karl Friston, Peter Kok, and Micha Heilbron for helpful discussion of our results. Additionally, we thank Varun Kapoor for the visualization of DNN features.

Views and opinions expressed are however those of the author(s) only and do not necessarily reflect those of the European Union or the European Research Council. Neither the European Union nor the granting authority can be held responsible for them.

## Author Contributions

**Conceptualization:** David Richter, Tim C. Kietzmann, Floris P. de Lange.

**Data curation:** David Richter.

**Formal analysis:** David Richter.

**Funding acquisition:** Tim C. Kietzmann, Floris P. de Lange.

**Investigation:** David Richter.

**Methodology:** David Richter, Tim C. Kietzmann, Floris P. de Lange.

**Project administration:** David Richter.

**Resources:** David Richter, Floris P. de Lange.

**Software:** David Richter.

**Supervision:** David Richter, Floris P. de Lange.

**Validation:** David Richter, Floris P. de Lange.

**Visualization:** David Richter.

**Writing – original draft:** David Richter.

**Writing – review & editing:** David Richter, Tim C. Kietzmann, Floris P. de Lange.

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
