## [Editor Report · Decision Letter 0]

10 Jan 2024

Dear Dr Richter, 

Happy New Year and apologies for the delay in getting back to you. We are still working on the backlog from the holidays.

Thank you for submitting your manuscript entitled "High-level prediction errors in low-level visual cortex" for consideration as a Research Article by PLOS Biology.

Your manuscript has now been evaluated by the PLOS Biology editorial staff as well as by an academic editor with relevant expertise and I am writing to let you know that we would like to send your submission out for external peer review.

Once your full submission is complete, your paper will undergo a series of checks in preparation for peer review. After your manuscript has passed the checks it will be sent out for review. To provide the metadata for your submission, please Login to Editorial Manager (https://www.editorialmanager.com/pbiology) within two working days, i.e. by Jan 12 2024 11:59PM.

Kind regards,

Christian

Christian Schnell, PhD

Senior Editor

PLOS Biology

cschnell@plos.org

---

## [Decision Letter · Decision Letter 1]

5 Mar 2024

Dear Dr Richter,

Thank you for your patience while your manuscript "High-level prediction errors in low-level visual cortex" was peer-reviewed at PLOS Biology. It has now been evaluated by the PLOS Biology editors, an Academic Editor with relevant expertise, and by several independent reviewers. 

In light of the reviews, which you will find at the end of this email, we would like to invite you to revise the work to thoroughly address the reviewers' reports.

As you will see below, the reviewers reviewers agree that the topic is interesting but that additional analyses are required to provide sufficient evidence for the claims. 

Given the extent of revision needed, we cannot make a decision about publication until we have seen the revised manuscript and your response to the reviewers' comments. Your revised manuscript is likely to be sent for further evaluation by all or a subset of the reviewers.

**IMPORTANT - SUBMITTING YOUR REVISION**

*Re-submission Checklist*

*Published Peer Review*

*PLOS Data Policy*

*Blot and Gel Data Policy*

Sincerely,

Christian

Christian Schnell, PhD

Senior Editor

PLOS Biology

cschnell@plos.org

REVIEWS:

Reviewer #1: As a disclaimer, I have previously reviewed this manuscript for another journal. I still have the same comments, which I attach below. 

In the manuscript entitled "High-level prediction errors in low-level visual cortex", Richter, Kietzmann and de Lange report an fMRI study in which they investigate the representational format of neural activity across several stages of visual processing, comparing conditions where expectations go unfulfilled and conditions in which no predictive context exists in a statistical learning paradigm. Specifically, they compare neural representational geometries to representational geometries obtained from different layers of a deep neural network. The main finding is that neural representational geometries during prediction errors resemble the representational spaces of deeper layers in the network (not earlier layers). This is consistent with early, intermediate, and high levels of neural processing assuming an abstract representational format. The study is statistically well powered, the analytical approach ingenious, and the results seem mostly robust. The manuscript thus constitutes and important addition to the literature on predictive processing by showing that results previously suggested by electrophysiology in non-human primates apply also to human visual processing and extend this literature by showing that these principles extend beyond the brain areas and stimuli tested in these studies. 

1. The finding that early layer representational spaces explain little to no variance during prediction errors is somewhat surprising. Although the finding is consistent with the overall hypothesis, I was wondering whether it is possible that the layer RDMs are highly correlated with each other, thus limiting the ability to partition their respective associated variance? Although the authors report that their image selection procedure minimized such correlations, the actual result correlations are not reported. It would be useful to provide evidence that the null results is not due to a problem with the predictors in the GLM. Variance inflation factors could be one useful criterion to determine whether there is an issue or not. 

2. The authors report that they used a decoding analysis to further refine the voxels within the otherwise anatomically delineated regions of interest. This decoding analysis was done on the basis of the localizer data. Several questions arise: 1. Was the same voxel selection procedure used to analyze the representational geometry of the localizer data? If so, the procedure seems potentially problematic because it might bias results towards finding the "feedforward" tuning properties; if not, comparisons between the prediction error and the localizer data may be invalid because of the different selection criteria. 2. When the ROI sizes were extended to account for the possibly confounding effects of voxel number, was this restricted to voxels that contributed to the decoding? 3. Did the representational geometry of the non-selected voxels in the respective ROIs resemble any of the DNN layers in the prediction error condition? This would give an indication as to whether the selectivity of neural populations factors into the results the authors report. 

3. Is it possible that the predictor letters drive the results, i.e., could the authors provide a control analysis in which they determine similarity between the predictor letters and apply that to the image-evoked activity? 

4. It is often recommended to z-transform correlation coefficients before subjecting them to t-tests. I suggest the authors try that here, too. 

Minor: 

- The methods section has several formulations that sound a bit weird in terms of language, e.g., in line 634 ("boilerplate"), 637 and 653 ("images were found"), 722 (presumably CSF and white matter time courses). 

- The reference to Behzadi et al. (line 677) seems to be missing. 

Reviewer #2 (Arjen Alink): This fMRI study is aimed at addressing the broadly relevant neuroscientific question of how prior knowledge affects the way in which our brain processes perceptual information. Specifically, the authors want to reveal whether the established expectation suppression (/surprise enhancement) effect in the early visual cortex (EVC) is driven by a mismatch in high- or low-level image features. To this end, they assessed whether enhanced EVC responses to unexpected images are predicted better by a mismatch in the visual features encoded in layer 2 vs layer 8 of a DNN (Alexnet, trained using ecoset). The key finding is that a mismatch in the visual features encoded in layer 8 best predicts univariate expectation suppression effects in EVC, while no such effect is observed for layer 2 features.

The topic and the conclusions drawn by this study are of sufficient general interest to be considered for publication in PLOS Biology. In its present form, however, the findings reported do not sufficiently support the main conclusions drawn by the authors, as pointed out in the main issues listed below. Resolving these issues would require additional analyses and/or a substantial 'toning-down' of the conclusions made. The latter could significantly affect the impact of the paper, and its suitability for PLOS Biology.

Major issue 1

The authors assume that layer 2 and layer 8 of their trained DNN represent low and high-level image features respectively. They, however, do not provide a compelling argument for this assumption, which is crucial to the main findings of this study. Based on Figure 3, it is impossible to say which layer best explains RDMs in EVC, LOC and HVC, as these areas are not highlighted. The authors should instead, or in addition, show how well each layer explains RDM characteristics for the EVC, LOC and HVC ROIs (using all voxels). If, e.g., layers 1 and 3 also robustly explain EVC RDMs, they need to be considered as well. Currently, the results presented (e.g., in 5a) look very convincing. However, it is not clear whether this is a result of the choice of layers. Therefore, I would strongly insist on the authors reporting their main result (Figure 5B and 5D) for all layers, at a minimum via a supplementary figure. 

Major issue 2

The main conclusion of the paper is reflected by the following sentence 544- "We find that visual prediction errors, a key feature of hierarchical PP, primarily reflect high-level visual

surprise, including in early visual cortex." This conclusion is driven by the observation that the magnitude of expectation suppression effects is best predicted by a mismatch in 'high-level visual features' encoded by layer 8, instead of the 'low-level visual features' of layer 2. However, the authors do not provide compelling support for their assumption that layer 2 measures low-level visual features while layer 8 encodes only high-level visual features. The authors show that surprise related to a mismatch in animacy nor a word-level (semantic) surprise can explain the observed relationship between 'layer 8 RDM surprise' and expectation suppression effects in the visual ROIs. However, they do not clarify what are the high-level layer 8 features that DO drive this effect.

To address this issue, the authors could, e.g., visualize the features (DNN filters, back projected to image space) of layer 8 (and for relevant additional layers, see major issue 1) that best explain expectation suppression effects in EVC. Here it is critical that the authors show that these features are more complex than low-level Gabor-like features that the EVC is thought to encode. 

Major issue 3

Multiple times, the authors suggest that the main effects reported could be related to more efficient processing of expected visual information in EVC (e.g., lines 398-400). Given these statements, I am surprised that the authors did not assess whether stimulus identity information in EVC, LOC and HVC response patterns increases for expected vs unexpected images, which would be consistent with de Lange's influential "less is more" finding. Such an analysis has to be performed and reported in this study, because it would allow readers to better understand how prediction error coding relates to 'efficient coding'. Also a null result, or an opposite-to-the-expected result, should be reported. 

Major issue 4

In the discussion, the authors acknowledge that attention and arousal effects cannot not be conclusively ruled out as an explanation for expectation effects. The authors, however, do not clearly highlight an aspect of the experimental paradigm that is of particular importance to this discussion.

Specifically, the task during expected trials 1) requires less visual attention and 2) involves a much simpler decision and motor preparation process as compared to unexpected trials. Hence, during expected trials participants can prepare a motor response as soon as they hear the sound cue, and then they simply execute it, as nothing unexpected happens. In contrast, during unexpected trials, participants will need to pause their prepared motor response, pay close attention to the image and then, possibly (~50/~50), abort their prepared response and initiate and execute a new motor response. Such differences between the main conditions appears to be consistent with the reported enhanced responses to unexpected image in the precentral gyrus. 

I agree that an attention/arousal/decision making effect does not provide a good explanation for the layer 8 surprise modulation for expectation suppression in EVC. However, two other arguments made against an attention-based explanation are not compelling. 

First of all, the authors find that expectation effects are stronger in earlier visual areas, and they argue that this rules out an attention-based explanation because attention effects should increase from lower to higher level brain areas. However, this argument hinges on two electrophysiological studies in animal models. I would like to stress here that the work of Logothetis (What we can do and what we cannot do with fMRI, 2008), has taught us that one cannot simply assume that spiking rate effects translate into comparable BOLD effects: "page 876- Yet, as I have indicated above, the BOLD signal is primarily affected by changes in excitation-inhibition balance, and this balance may be controlled by neuromodulation more than by the changes in spiking rate of a small set of neurons. In fact, the BOLD signal is strongly modulated by attention…".

Second, the argument that an expectation-based explanation is more parsimonious is inconsistent with the recent claim by Richter and de Lange that expectation effects are gated by attention (D Richter, FP de Lange - eLife, 2019). Hence, given the authors previous arguments and findings, an expectation-based explanation of expectation suppression requires an attention component, and therefore a pure and parsimonious expectation-based explanation does not exist.

Other issues:

- In this paper reaction times are shown to increase with layer 8 surprise. It is unclear, however, whether or not this is related to category effects (or semantic effects). For completeness, it would be good if the authors report this.

- in Figures 3 and 5, layers 1-8 colours should not be presented on a continuous colour scale, as it is a discrete variable (there is no layer 2.5). A single colour block per layer would make it much easier to find out which blob relates to which layer 

- Figure 4: the lower figure depicting the main effect (scatter plot) is very difficult to understand because: the x-axis is not labelled (z-scored corr. dist), the y-axis is not labelled (unexp-exp modulation). Moreover, the authors should explain in the legend why the dots are arranged in lines (resolution of the RDMs?) 

- Related to the above point, the authors perform linear regression for their main analysis of layer8-exp suppression modulation. However, based on the 4b plot, it is not possible to see whether the linearity assumption is valid. The authors should statistically assess whether their linearity assumption is justified. 

Reviewer #3: General comment:

1) Expectations, or predictions, influence our perception, but the underlying neural mechanisms are unclear. This study investigates how prediction errors are processed across visual cortex, using FMRI and representational distances based on a deep neural network (DNN). It is reported that high-level prediction errors are found in early visual cortex, and argued to better fit proposals in which prediction error processing reflects higher-order cortical rather than local feature representations, consistent with recent work in non-human primates (e.g., Uran et al., 2022 Neuron). The FMRI and computational methods appear solid, but I think a number of questions should be addressed, to clarify interpretations of the results, related to how low-level and high-level features have been defined, the behavior and FMRI data in the prediction-free condition. Please see specific comments below.

Specific comments:

2) The definition of low-level vs high-level features were derived from DNN layers. However, it is not clear from the stimulus examples given (Fig. 4A), to what degree that low-level and high-level features defined in this way correspond to the common usage of these terms in vision research. For example, "high-level" included quite detailed (low-level) features in the examples, and "low-level" seems to carry overall gist or object identification (high-level). Perhaps this could be clarified in the main text?

3) Further, in Fig. 4B, the building versus hand differ markedly in low-level features, seemingly more than the difference in guitars. However, if I am interpreting the color-code correctly, the representational dissimilarity matrix (RDM) for low-level features shows a considerably smaller difference for building versus hand (compared with the guitars). So what is the DNN picking up on?

4) The large high-level surprise example in Fig. 4A differs in both high- and low-level features compared with the expected object. To help with controlling for differences at multiple levels, were stimuli used where low-level features were matched but only high-level features differed? It would be useful to present cases of such stimuli. 

5) RE: Fig. 5. Is it possible that layer 8 may include high-level surprise as well as low-level surprise elements (considering the example images provided in the paper)? That is, is it possible, for instance, that layer 2 includes low-level surprise and layer 8 includes low-level and high-level surprise, which might lead to a greater prediction error overall for the latter? Could this contribute to the differences seen in Fig. 5B, where V1 shows the highest modulation for layer 8 (responding to both high and low levels), whereas LOC and HVC show less modulation (responding only to high levels)?

6) RE: Fig. 3. As currently presented, it appears that early visual cortex may correspond to DNN layers 1-4-ish and higher visual cortex to DNN layers 5-8-ish. So it would be useful to show ROIs in addition to the current DNN layer color code, to be able to more specifically gauge how DNN layers relate to the ROIs.

7) RE: Fig. 7A. To further substantiate the importance of high-level surprise, it would be useful to show reaction time modulation based on each of the DNN layers (not just layers 2 and 8)?

8) RE: (From the abstract) "…prediction errors may be computed at higher levels and propagate down the cortical hierarchy". It seems elsewhere in the paper, the idea that comes across (at least to me) is that high-level predictions are transmitted down to early visual cortex and the high-level errors are generated at this early stage. Perhaps this could be clarified in the text.

9) RE: scatter plots in Fig. 4B and 5D. (I may have missed it in the text but) Perhaps add an R value to help the reader assess the effect.

---

## [Decision Letter · Decision Letter 2]

8 Aug 2024

Dear Dr Richter,

Thank you for your patience while we considered your revised manuscript "High-level prediction errors in low-level visual cortex" for publication as a Research Article at PLOS Biology. This revised version of your manuscript has been evaluated by the PLOS Biology editors, the Academic Editor and the original reviewers.

Based on the reviews and on our Academic Editor's assessment of your revision, we are likely to accept this manuscript for publication, provided you satisfactorily address the remaining points raised by the reviewers and the following data and other policy-related requests.

* We would like to suggest a different title to improve accessibility for our broad readership: "High-level predictions in the visual cortices constrain sensory processing in earlier visual areas"

* Please include information about whether the study has been conducted according to the principles expressed in the Declaration of Helsinki.

* DATA POLICY:

Regardless of the method selected, please ensure that you provide the individual numerical values that underlie the summary data displayed in the following figure panels as they are essential for readers to assess your analysis and to reproduce it: 2AB, 5B, 6B, S1B, S2, S4A, S5 and S7AB

* CODE POLICY

* Please note that per journal policy, the model system/species studied should be clearly stated in the abstract of your manuscript. 

We expect to receive your revised manuscript within two weeks. 

*Published Peer Review History*

*Press*

Sincerely,

Christian

Christian Schnell, PhD, 

Senior Editor

cschnell@plos.org

PLOS Biology

Reviewer remarks:

Reviewer #1: The authors have addressed all points I had regarding their original submission. 

Reviewer #2 (Arjen Alink): I would like to thank the authors for addressing all the issues I raised in such a thorough manner. With these changes made to the manuscript, I fully support its publication in the journal of PLOS Biology

Reviewer #3: I appreciate the effort of the authors in running the additional analyses as requested by the reviewers. Some of these additional analyses provided important information to interpret the results, but were not included in the manuscript. I recommend that they be included in the manuscript (not just the response to reviewers), to increase transparency; and it would be a quick revision as the analyses have already been done and could simply be cut and pasted into the manuscript. That is, I suggest the following to be added to the manuscript.

1) Please include the figure: "Representational similarity: DNN and visual stimulation" as a supplementary figure in the manuscript, which seems to only be contained in the Response to Reviewers. I think this figure is very useful to allow the reader to interpret the main results of the paper, e.g., it shows an overlap between early layers and the HVC. 

2) I suggest including a statement in the manuscript, clarifying that layers of visual DNNs tend to highly correlate so "ROI analysis results assessing the modulation of visual responses by surprise as indexed by feature dissimilarity across all 8 layers of the DNN" do not show reliable results. (RE: "Figure depicting the ROI analysis results assessing the modulation of visual responses by surprise as indexed by feature dissimilarity across all 8 layers of the DNN. ROIs in early visual (V1), intermediate (LOC) and higher visual cortex (HVC), show no reliable results. Error bars indicate the 95% within-subject confidence intervals. Gray dots denote individual subjects.") This is important to allow the reader to assess the limits of the approach.

3) Please include the following results in the manuscript which show that expected vs unexpected stimuli could not be decoded from the ROIs (which is surprising considering the main claims in the paper), i.e., (line 605 in response to reviewers) "…to obtain a mean decoding accuracy for expected and unexpected trials per participant. On average decoding of the object stimuli was well above chance (12.5%), but did not reliably differ between expected and unexpected stimuli in any of the three ROIs: HVC (Expected: 47.5%, ±3.0%; Unexpected: 46.9%, ±3.0% [mean, ±95%CI]), LOC (Expected: 63.1%, ±2.2%; Unexpected: 64.5%, ±2.6%), V1 (Expected: 73.1%, ±3.3%; Unexpected: 73.8%, ±3.3%).

ROI Test statistic P value

V1 t(32) = -0.51 p = 0.613

LOC t(32) = -1.02 p = 0.315

HVC t(32) = 0.55 p = 0.588

---

## [Editor Report · Decision Letter 3]

3 Sep 2024

Dear David,

Thank you for the submission of your revised Research Article "High-level visual prediction errors in early visual cortex" for publication in PLOS Biology. On behalf of my colleagues and the Academic Editor, Caspar Schwiedrzik, I am pleased to say that we can in principle accept your manuscript for publication, provided you address any remaining formatting and reporting issues. These will be detailed in an email you should receive within 2-3 business days from our colleagues in the journal operations team; no action is required from you until then. Please note that we will not be able to formally accept your manuscript and schedule it for publication until you have completed any requested changes.

PRESS

Sincerely, 

Christian

Christian Schnell, PhD

Senior Editor

PLOS Biology

cschnell@plos.org